# Evaluating Robustness of Reasoning Models on Parameterized Logical Problems

**Naïm Es-sebbani** [1]    **Esteban Marquer** [2]    **Yakoub Salhi** [1]    **Zied Bouraoui** [1]

## Abstract

Logic provides a controlled testbed for evaluating LLM-based reasoners, yet standard SAT-style benchmarks often conflate surface difficulty (length, wording, clause order) with the structural phenomena that actually determine satisfiability. We introduce a diagnostic benchmark for *2-SAT* built from parameterized families of structured 2–CNF formulas, where satisfiability is characterized by the implication graph and can be tuned along interpretable axes. Our generators isolate distinct competencies and failure modes: (i) contradiction-cycle UNSAT cores with controllable size and imbalance, (ii) SAT instances with a prescribed fraction of free variables to control solution multiplicity, (iii) planted backbones that modulate propagation, (iv) late bridge clauses that couple otherwise monotone regions to probe sensitivity to ordering and revision, and (v) symmetry/duplication variants that test abstraction under renaming and redundant structure. We evaluate LLM-based reasoners on decision accuracy and assignment validity, and quantify robustness under semantics-preserving perturbations such as clause reordering, filler clauses, and variable renaming. Across models, we observe sharp performance transitions under targeted structural interventions even when instance size or satisfiability is held fixed, revealing brittleness regimes that are invisible to aggregate SAT accuracy. [1]

## 1. Introduction

Large reasoning models (LRMs) augment standard language models with training- and inference-time mecha-nisms that encourage explicit multi-step computation, and have achieved strong empirical performance on several domains, including mathematical reasoning and code generation (Patel et al., 2023; Bubeck et al., 2023). Motivated by these gains, recent benchmarks increasingly emphasize combinatorial search and constraint satisfaction, including planning and propositional satisfiability (Wei et al., 2025). While LRMs often outperform conventional LLMs in these settings, recent SAT-based evaluations show that many models remain sensitive to instance hardness, with performance dropping near phase-transition regions; some stronger LRMs are more robust, but still exhibit failures in harder or less familiar formats (Hazra et al., 2025). One interpretation is that some LRMs approximate solver-like procedures through behaviors such as heuristic variable selection, unit propagation, backtracking, self-correction, and candidate selection (Hazra et al., 2025). If such behavior reflects transferable structural reasoning rather than format-specific pattern reuse, then performance should be relatively stable under semantics-preserving transformations and should degrade primarily with controlled changes in structural difficulty.

A central question is therefore how to distinguish transferable algorithmic competence from task-specific pattern reuse. A growing body of evidence points to brittle generalization: models can fail even when instances admit short solutions (Ariyani et al., 2025), exhibit systematic errors on certain relational reasoning classes (Das et al., 2025), and transfer poorly across closely related maze distributions (Valmeekam et al., 2025a). Prior work on Long-CoT distillation further suggests that the global structure of reasoning demonstrations can matter more than the semantic correctness of individual intermediate steps: models remain robust to local content corruptions, but degrade when reasoning steps are shuffled, deleted, or otherwise made structurally incoherent (Li et al., 2025). These findings motivate evaluations that separate structure tracking from reliance on familiar trace formats, local heuristics, and presentation-specific cues.

We study this question through a diagnostic evaluation regime based on parameterized problem families. The goal is not to measure worst-case SAT solving ability, but to avoid the common confound in aggregate SAT-style benchmarks, where instance size, surface presentation, and structural

---

[1] CRIL UMR 8188, Univ Artois, CNRS, France   [2] GREYC, Université de Caen Basse Normandie, France. Correspondence to: Naïmhe Es-sebbani <essebbani@cril.fr>.

*Proceedings of the $43^{rd}$ International Conference on Machine Learning*, Seoul, South Korea. PMLR 306, 2026. Copyright 2026 by the author(s).

[1] The dataset, source code, and prompts are available at https://github.com/essebbaninaim/reasoning-robustness-llm.

phenomena can vary together. We instead vary difficulty along explicit structural axes while keeping the verification objective fixed. Concretely, we focus on satisfiability of 2-CNF formulas. Although 2-SAT is polynomial-time solvable, this tractability is a feature of our design: it avoids confounding logical reasoning failures with the absence of an efficient search strategy. At the same time, 2-SAT has a rich and interpretable structural theory. Its implication graph and strongly connected components provide a direct link between graph-theoretic properties and satisfiability. This gives a principled way to manipulate structural phenomena, such as contradiction cores and propagation paths, without changing the underlying verification problem. It also enables controlled instance generation, solver-certified labels and witnesses, and semantics-preserving perturbations.

We design generators that target distinct competencies, including long-range implication tracking, localization of contradiction cores, backbone propagation, and invariance to clause order, redundant clauses, and variable renaming. This fine-grained control enables stress tests beyond aggregate SAT/UNSAT accuracy and reveals brittleness regimes that are not visible under a single instance distribution. Accordingly, our claims are intentionally scoped: the benchmark probes presentation-invariant structural reasoning on parameterized 2-CNF families, rather than general logical reasoning in full.

Our contributions are threefold. First, we introduce a diagnostic 2-SAT benchmark built from parameterized families of structured 2-CNF formulas with controllable implication-graph signatures and semantics-preserving perturbations. Second, we propose an evaluation suite that goes beyond aggregate SAT/UNSAT accuracy by checking both decision correctness and assignment validity, thereby distinguishing correct classification from the ability to construct a valid satisfying witness. We also quantify robustness under targeted interventions such as clause reordering, filler clauses, and variable renaming. Third, we present a systematic empirical study of LLM-based reasoners across these families. Empirically, we find sharp performance transitions and generator-specific brittleness even when instance size is held fixed. We interpret these results as evidence that, on the parameterized 2-CNF families studied here, current LLM-based reasoners do not consistently implement presentation-invariant structural reasoning.

## 2. Related work

**Reasoning in LLMs and LRMs.** A large literature evaluates language models as reasoners across inference regimes, including deductive (Clark et al., 2020; Saeed et al., 2021), abductive (Kazemi et al., 2023), commonsense (Tian et al., 2021), and logical or symbolic reasoning (Jiang et al., 2024). Beyond language-centric benchmarks, recent work probes

models on algorithmic and combinatorial tasks that more directly instantiate search and constraint satisfaction, including graph search (Rameshkumar et al., 2025), planning (Valmeekam et al., 2025b), puzzles (Shojaee et al., 2025), propositional satisfiability (Hazra et al., 2025), and graph coloring (Heyman & Zylberberg, 2025). Across these settings, the required inference depth is often a strong predictor of difficulty (Parmar et al., 2024). Methodologically, three common approaches are used to elicit or support reasoning: prompting-based scaffolds such as CoT (Kojima et al., 2022; Saparov & He, 2023) and in-context exemplars (Wang et al., 2023); specialized reasoning models that improve over fine-tuning (Valmeekam et al., 2024); and tool-oriented pipelines that translate problems into symbolic forms executed by external solvers or verifiers (Jiang et al., 2024).

**Logical benchmarks and SAT-based evaluation.** A prominent line of work studies formal inference when logical statements are rendered in simplified natural language. Clark et al. (2020) report near-perfect accuracy on Horn-rule entailment in such settings, but subsequent analyses find brittleness under distribution shift, consistent with shortcut learning rather than systematic rule-based generalization (Zhang et al., 2023). Related observations hold for satisfiability-style evaluation: Richardson & Sabharwal (2022) consider hard propositional SAT instances presented in simplified language and highlight sensitivity to training data selection and limited extrapolation to larger instances. Another approach seeks robustness by eliciting explicit proof traces, relying on the premise that reliable local steps can compose into longer derivations (Saha et al., 2020; Tafjord et al., 2021; Creswell et al., 2023).

**Logical reasoning datasets.** Benchmarks such as LogiQA (Liu et al., 2020), ReClor (Yu et al., 2020), BoardgameQA (Kazemi et al., 2023), and CLUTRR (Sinha et al., 2019) probe logical and relational reasoning in text, but often entangle formal inference with world knowledge and linguistic priors. In contrast, datasets such as FOLIO (Han et al., 2024a), RuleTaker (Clark et al., 2020), and P-FOLIO (Han et al., 2024b) more directly target knowledge-light logical inference. Logical puzzle corpora provide another controlled avenue (Giadikiaroglou et al., 2024), including ZebraLogic (Lin et al., 2025) and AutoLogi (Zhu et al., 2025). Finally, some recent benchmarks evaluate models on SAT-style inputs more directly, e.g., PARAT (Pan et al., 2025), SATBench (Wei et al., 2025), and SATQuest (Zhao et al., 2026), which frame satisfiability problems as natural language puzzles and leverage classical SAT solvers for verification.

**Position.** Compared to prior logic and SAT benchmarks, we emphasize *parameterized* instance families with fine-grained structural control and *semantics-preserving perturbations* (e.g., clause order, filler clauses, and renaming

invariances). This differs from SAT-in-natural-language settings that primarily vary surface realizations of fixed instances, and from proof/trace-based evaluations that require models to emit (and be graded on) explicit derivations. Instead, we keep the verification objective unchanged and probe whether models' decisions and assignments remain stable under transformations that preserve satisfiability while targeting specific structural competencies. Focusing on 2-SAT enables mechanistic instance design because satisfiability is characterized by the implication graph, allowing attribution of failures to concrete graph phenomena rather than to instance size or surface form alone.

## 3. Problem setting

We study the satisfiability of 2–CNF formulas and design parameterized instance families whose difficulty can be tuned through interpretable structural properties. Throughout, let $\mathcal{V} = \{x_1, \ldots, x_n\}$ be a finite set of Boolean variables, and let $\mathcal{L} = \mathcal{V} \cup \{\neg x \mid x \in \mathcal{V}\}$ be its associated set of literals. Given a literal $\ell$, we write $\overline{\ell}$ for its complementary literal. We use the usual definitions $\ell \to \ell' := \overline{\ell} \lor \ell'$ and $(\ell \leftrightarrow \ell') := (\ell \to \ell') \land (\ell' \to \ell) \equiv (\overline{\ell} \lor \ell') \land (\overline{\ell'} \lor \ell)$.

A *2–CNF* is a conjunction of clauses, each clause a disjunction of at most two literals. A literal in a 2–CNF formula is *pure* if it appears with only one polarity. $n := |\mathcal{V}|$ denotes the number of variables and $m := |\mathcal{C}|$ the number of clauses.

**Implication graph for 2-SAT.** To decide satisfiability of a 2–CNF formula, we associate with it an implication graph on vertex set $\mathcal{L}$. Each clause $(\ell_1 \lor \ell_2)$ contributes the edges $\overline{\ell_1} \to \ell_2$ and $\overline{\ell_2} \to \ell_1$. We use $l \rightsquigarrow l'$ to denote that there is a path from $l$ to $l'$ in the implication graph.

A 2–CNF formula is unsatisfiable (UNSAT) iff there exists a variable $x$ such that $x$ and $\neg x$ belong to the same Strongly Connected Component (SCC) of the implication graph (equivalently, $x \rightsquigarrow \neg x$ and $\neg x \rightsquigarrow x$).

**Why 2-SAT?** Our choice of 2-SAT is motivated by controllability rather than worst-case hardness. The implication-graph characterization gives a direct and interpretable mapping from graph structure to satisfiability, allowing us to vary structural properties while keeping formula size and the verification objective controlled. This is less direct in other tractable fragments such as Horn-SAT, where satisfiability is determined by forward chaining and does not provide the same SCC-based decomposition used by our generators. Moving to harder classes such as 3-SAT would introduce a different confound: failures could reflect either weak structural reasoning or the absence of an efficient search strategy. By contrast, 2-SAT lets us focus on structural tracking and invariance under controlled perturbations.

**Difficulty controls.** For each generator, we vary the instance

structure and its presentation (clause order, variable renaming), which does not change satisfiability but can stress heuristic sensitivity. For UNSAT generators, we mainly vary (i) the structure of the implication-graph witness of unsatisfiability (the *UNSAT core*) and (ii) the number of *filler* clauses, i.e., clauses that can be satisfied independently of the core and thus do not contribute to the contradiction. For the two SAT generators that parameterise *critical variables*, we vary (i) the number of critical variables, namely backbone variables in the Backbone generator and free variables in the EquivalenceCore generator, (ii) the total number of variables, and (iii) the number of pure literals (literals that occur with only one polarity and hence do not constrain satisfiability). The remaining two generators (ImplicationCycle and MonoBridge) do not parameterise critical variables and instead control difficulty via structural parameters of the implication graph, namely the cycle split position and the bridge clause placement, respectively.

### 3.1. UNSAT Generator using Implication Cycles

We pick distinct variables $v_1, \ldots, v_L$ and for each we pick a literal $\ell_i \in \{v_i, \neg v_i\}$ for $i = 1, \ldots, L$. Then, we construct a directed cycle in the implication graph:

**Forward part:** $\quad \ell_1 \to \ell_2 \to \cdots \to \ell_k \to \overline{\ell_1},$

**Backward part:** $\quad \overline{\ell_1} \to \ell_{k+1} \to \cdots \to \ell_L \to \ell_1.$

Encoding each implication $\ell \to \ell'$ as the single 2-clause $\overline{\ell} \lor \ell'$ yields a core that forces both $\ell_1 \to \overline{\ell_1}$ and $\overline{\ell_1} \to \ell_1$, hence UNSAT. The construction above contains one clause per displayed implication. The forward part contributes $k$ clauses; the backward part contributes $(L - k + 1)$ clauses, for a total of $L+1$ clauses.

**Parameters controlling difficulty.** Difficulty is controlled by four parameters: (i) the *cycle split* $(L, k)$, where $L$ is the number of literals in the core cycle and $k$ is the position of the twist to $\overline{\ell_1}$ (equivalently, the *imbalance* $k/L$, with extremes $k = 1$ and $k = L$); (ii) the number of *filler clauses* $r$ that are satisfiable independently of the core (controlling size/density); (iii) the *presentation*, i.e., whether core clauses are ordered along the implication chain or shuffled and interleaved with fillers; and (iv) the *total number of variables* $n \geq L$, with variables outside the core appearing only in fillers (controlling the ratio $L/n$). The key structural difficulty is the imbalance $k/L$: when it is close to 0 or 1 (e.g., $k = 1$ or $k = L$), one direction of the contradiction cycle is long while the other is short, forcing long-range implication tracking before closing the cycle.

### 3.2. SAT Generator via Free Variables

We choose a set $F \subseteq \mathcal{V}$ of *free variables* with $|F| = f$, meaning that the values of variables in $F$ can be chosen arbitrarily. For every remaining variable $y \in \mathcal{V} \setminus F$, we choose

a parent $p(y) \in F$ and a literal $\tau(y) \in \{p(y), \neg p(y)\}$, and enforce $y \leftrightarrow \tau(y)$. Each such definition contributes two binary clauses (two implications). Every assignment to $F$ extends uniquely to a satisfying assignment on all of $\mathcal{V}$.

**Parameters controlling difficulty.** The number of variables $n$ and free variables $f$ determines the model count $2^f$. The *free-to-bound ratio* $f/(n-f)$ controls coupling strength: smaller ratios induce stronger propagation from a small free set. We also vary the *presentation order*, placing definition clauses early or late in the clause list to probe order sensitivity. The formula has $2(n-f)$ clauses. This generator isolates sensitivity to solution multiplicity, through $f$, and to the amount of propagation required to produce a valid assignment.

### 3.3. Backbone-based Method

The goal is to construct a satisfiable 2–CNF that is the conjunction of: (i) clauses that infer a planted *backbone* assignment, i.e., an assignment on a subset of variables that is forced in every satisfying assignment (these variables take the same truth value in all solutions), and (ii) a remainder consisting only of monotone 2-clauses (all literals have the same polarity). We choose a backbone $B \subseteq \mathcal{V}$ with a planted assignment $b : B \to \{0,1\}$, and let $F = \mathcal{V} \setminus B$ be the free part. We introduce a fresh auxiliary variable $a_x$ for each $x \in B$. The formula is a conjunction of two parts:

$$\Phi^\star := \underbrace{\Phi_{\text{backbone}}}_{\text{infers } b \text{ using only 2-clauses}} \wedge \underbrace{\Phi_{\text{mono}}}_{\text{monotone 2-clauses over } F} .$$

**(A) Backbone inference.** For every $x \in B$, we add the following clauses using a fresh auxiliary $a_x$:

Case $b(x) = 1$: $(x \vee a_x) \wedge (x \vee \neg a_x)$,

Case $b(x) = 0$: $(\neg x \vee a_x) \wedge (\neg x \vee \neg a_x)$.

Each pair is logically equivalent to the unit clause it emulates (respectively $x$ or $\neg x$), but uses only binary clauses; thus it fixes $x$ to $b(x)$ in all satisfying assignments.

**(B) Monotone remainder.** We choose an orientation $\pi \in \{+, -\}$ and add only 2-clauses over variables in $F$:

Case $\pi = -$ : add clauses $(\neg y \vee \neg z)$ for $y, z \in F$,

Case $\pi = +$ : add clauses $(y \vee z)$ for $y, z \in F$.

No variable from $B$ appears in $\Phi_{\text{mono}}$. If $\pi = -$, setting all $y \in F$ to 0 satisfies $\Phi_{\text{mono}}$; if $\pi = +$, setting all $y \in F$ to 1 does. Together with the pinned backbone, $\Phi^\star$ is satisfiable.

**Parameters controlling difficulty.** Let $|B| = \beta n$ and $m_{\text{mono}}$ be the number of monotone clauses added over $F$. The part $\Phi_{\text{backbone}}$ has exactly $2|B|$ clauses and $|B|$ auxiliaries. The total clause count is $|\mathcal{C}| = 2|B| + m_{\text{mono}}$. Difficulty is tuned via the *backbone fraction* $\beta$, *monotone density* relative to possible pairs $\frac{2m_{\text{mono}}}{|F|^2}$, and *orientation* $\pi$.

### 3.4. Monotone Split with Moving Cross-Polarity Bridge

We construct a 2–CNF that consists of (i) a positive-monotone part over variables $P$ (only clauses of the form $(p \vee p')$ with $p, p' \in P$), (ii) a negative-monotone part over variables $N$ (only clauses of the form $(\neg q \vee \neg q')$ with $q, q' \in N$), with $P \cap N = \varnothing$, and (iii) a single bridge clause that couples both parts $(\neg p^\star \vee q^\star)$ with $p^\star \in P$ and $q^\star \in N$. The bridge is inserted at a chosen position in the clause list so we can probe when a solving method backtracks as it scans clauses.

**Parameters controlling difficulty.** The number of clauses is $|\mathcal{C}| = m_+ + m_- + 1$ with $m_+$ and $m_-$ the number of positive and negative clauses, plus the bridge. Densities $m_+/|P|$ and $m_-/|N|$ and the proportion of each part control how strongly each monotone part biases decisions. *Bridge position* $s$ probes sensitivity to late coupling and potential revision during clause scanning; satisfiability is invariant under $s$. In particular, we parametrize generation with the *overall variable density* $\frac{|P|+|N|}{2|\mathcal{C}|}$, the relative size of the positive part $m_+/|\mathcal{C}|$, and the position $s \in [1, |\mathcal{C}| - 1]$ of the bridge among the clauses.

### 3.5. Symmetry/Redundancy Probe

Given a base 2–CNF $\Psi$ over variables $\mathcal{V}$, we build a new instance $\Phi := \Psi \wedge \rho(\Psi)$, where $\rho$ is a bijective renaming from $\mathcal{V}$ to a fresh, disjoint set $\mathcal{V}'$ (so $\mathcal{V} \cap \mathcal{V}' = \varnothing$). Thus $\Phi$ is the concatenation of two isomorphic, independent copies of $\Psi$. Comparing a model on $\Psi$ versus on $\Phi$ probes whether the model detects the repeated structure and how its behavior is affected by it. We do not assume that a model should exploit the symmetry to reduce computation, since doing so requires identifying the correspondence between the two copies.

**Properties (independence and symmetry).** *Independence.* $\mathcal{V}$ and $\mathcal{V}'$ are disjoint, so there is no logical interaction. *Automorphism.* Swapping $\mathcal{V} \leftrightarrow \mathcal{V}'$ (i.e. applying $\rho$ or $\rho^{-1}$) leaves $\Phi$ invariant. $\Phi$ has a nontrivial symmetry. *Satisfiability.* $\Phi$ is SAT iff $\Psi$ is SAT. If $\Psi$ has $M$ models, then $\Phi$ has $M^2$ models (Cartesian product across copies). If $\Psi$ is UN-SAT, so is $\Phi$. We therefore use this construction to measure sensitivity to repeated structure, e.g., whether duplication degrades accuracy and whether reasoning traces show any cross-referencing between the two copies, rather than as a test of computational efficiency.

**Variants and extensions.** *More copies.* We use $d > 2$ disjoint renamings to form $\Psi \wedge \rho_2(\Psi) \wedge \cdots \wedge \rho_d(\Psi)$, probing scaling under repeated symmetry. *Copy-wise shuffles.* Randomly permute the clause order within each copy; or apply a variable permutation per copy to avoid accidental alignment in indices. *Trigger placement.* If $\Psi$ has a late trigger clause that forces a contradiction, we place its renamed counterpart

early (or vice versa) to study sensitivity across copies.

**Scope of renaming** The renaming probed in this section is *cross-copy* renaming: the second copy uses a variable set disjoint from the first, so models see the same logical content under entirely fresh symbols. A second, complementary form of renaming is exercised across the six template verbalizers (Section G): the same underlying variable is mapped to different entity types, including letters, person names, room numbers, and team labels, yielding entity-level renaming at a fixed clause structure. We do *not* test *within-instance* variable renaming, such as permuting variable indices while holding template, clause order, and other surface attributes fixed. Such a probe would isolate sensitivity to the surface identity of individual variables rather than to their structural role; we leave this finer-grained test to future work.

## 4. Dataset Construction

**Verbalizers.** We convert each 2–CNF into text using two approaches as illustrated in Fig 10. The *template verbalizer* deterministically renders each clause into a short natural-language statement, using a fixed mapping from variables to entity names within a theme. We use 6 different templates to reformulate the SAT problem into a more practical problem: a direct verbalization of how the clause reads (logic), letter passing between people (letter), assignment of people to a blue and a red team (team), affinities between invitees to constrain a social gathering (social), room lighting rules (room), door locking (door). Each template maps a clause to a fixed sentence pattern with named entities (e.g., $\neg a \vee b$ becomes "either a is on the Blue team or b is on the Red team"). The mapping is deterministic and invertible, so the logical content of each clause is preserved by construction.

The **LLM verbalizer** uses an LLM to rewrite each clause as a short narrative paragraph. We present the LLM with the two literals as natural-language statements derived from a thematic entity mapping (e.g., "the butler is innocent", "the detective is the murderer") and ask it to produce a paragraph expressing their disjunction. We use three themes (spy, heist, detective) to guide stylistic context. To prevent information loss, each paragraph is checked by a two-step LLM validator: it (i) extracts the two referenced entities and (ii) infers whether each is stated positively or negatively. We compare the recovered clause to the original; mismatches trigger regeneration (up to three attempts), after which we fall back to a deterministic template. On the augmented dataset, 95.4% of clauses validate on the first attempt and only 3.3% use the fallback, indicating that the verbalizer is typically semantically faithful.

**Choice of generation parameters.** To generate the dataset, we use the parameters described in Section 3 for each generator. To facilitate sampling, each parameter is designed to be in $[0, 1]$. In our experiments, for each parameter, we split its possible values into thirds and generate data using an easy to manipulate value close to the center of each interval of the parameter space ($0.2 \in [0; \frac{1}{3}]$, $0.5 \in [\frac{1}{3}, \frac{2}{3}]$, $0.8 \in [\frac{2}{3}, 1]$). When studying the impact of each parameter, we considered 4 additional values that correspond to the bounds of those intervals ($0, \frac{1}{3}, \frac{2}{3}, 1$). For generators where more than one major difficulty parameter is identified, we vary only one parameter at a time and set the others to $0.5$ to minimize the number of settings to evaluate. When generating data, the number of generated clauses is the main generation parameter, but minor differences might appear depending on the specifics of each generator. For instance, some generators are unable to generate an odd number of core clauses. Hereafter, the number of clauses is given with a $\pm 1$ tolerance to best match the target difficulty.

**Pipeline.** For a given generator parameter combination, we generate 10 formulas and produce, for each of them, either 6 verbalizations for the template verbalizer or 3 for the LLM verbalizer (only in the ablation study Section 5.5).

## 5. Experiments

Table 1 reports results for seven public reasoning models (14B to 120B) across the four main generators, with all generator-specific difficulty parameters fixed to $0.5$ and only the number of clauses varied. Model details appear in Appendix Table 3. For each setting, we sample 10 formulas and generate 6 template verbalizations per formula (60 prompts). We parse the SAT/UNSAT decision and, when the model predicts SAT, an assignment, counting truncated or unparsable outputs as incorrect. Ablations focus on four models (Phi4-reasoning, Phi4-reasoning-plus, Qwen3-Next-80B, GPT-OSS-120B) under the same protocol unless noted. Beyond this open-weight evaluation, we also report (i) per-(generator, $|\mathcal{C}|$) results for three proprietary reasoning models (o3-mini, o4-mini, GPT-5.4-mini) in Section N, (ii) an ablation of GPT-OSS-120B across the `low`/`medium`/`high` reasoning-effort in Section O, and (iii) a tool-calling variant of GPT-5.4-mini with `PySAT` code execution in Section P.

**Number of clauses as the main difficulty parameter.** Increasing $|\mathcal{C}|$ simultaneously increases (i) input length and parsing burden, and (ii) the amount of structure the model must integrate to reach a consistent global conclusion. We therefore treat $|\mathcal{C}|$ as the primary scaling axis. Importantly, $|\mathcal{C}|$ conflates core clauses (that witness SAT/UNSAT) with non-core clauses; in section 5.2 we decouple these to test whether failures are driven by reasoning depth in the core or by distractor load. Phi4-reasoning-plus frequently produces outputs that exceed our maximum output budget, leading to truncated or unparsable answers even on moderate sizes. We report these cases as incorrect (per the protocol above). Truncation rates are in Appendix Table 4.

| Generator | Model $|\mathcal{C}|$ | Llama-3.3 70B-Instruct Sat. | Wit. | OLMo-3 32B Sat. | Wit. | Phi4 reasoning Sat. | Wit. | Phi4 reasoning-plus Sat. | Wit. | QwQ 32B Sat. | Wit. | Qwen3-Next 80B Sat. | Wit. | GPT-OSS 120B Sat. | Wit. |
|---|---|---|---|---|---|---|---|---|---|---|---|---|---|---|---|
| ImplicationCycle | 5 | 0.0 | — | *98.3* | — | **83.3** | — | 95.0 | — | *98.3* | — | **100.0** | — | *91.7* | — |
| | 10 | 1.7 | — | 95.0 | — | 83.3 | — | 75.0 | — | *96.7* | — | **98.3** | — | 95.0 | — |
| | 15 | 15.0 | — | **96.7** | — | 61.7 | — | 40.0 | — | 68.3 | — | **96.7** | — | *93.3* | — |
| | 20 | 18.3 | — | **88.3** | — | 41.7 | — | 15.0 | — | 65.0 | — | *96.7* | — | *93.3* | — |
| | 50 | 11.7 | — | 65.0 | — | 3.3 | — | 0.0 | — | 40.0 | — | **81.7** | — | *56.7* | — |
| | 75 | 13.3 | — | *50.0* | — | 1.7 | — | 0.0 | — | 48.3 | — | **73.3** | — | 45.0 | — |
| | 100 | 20.0 | — | *51.7* | — | 0.0 | — | 0.0 | — | 45.0 | — | **76.7** | — | 40.0 | — |
| EquivalenceCore | 10 | **100.0** | 1.7 | **100.0** | *93.3* | **100.0** | 80.0 | **100.0** | 86.7 | **100.0** | 86.7 | **100.0** | **91.7** | **100.0** | 83.3 |
| | 15 | *98.3* | 0.0 | **100.0** | **55.0** | 90.0 | 35.0 | 95.0 | 35.0 | *96.7* | 45.0 | **100.0** | *53.3* | **100.0** | 46.7 |
| | 20 | 96.7 | 0.0 | 91.7 | 56.7 | 86.7 | 21.7 | 83.3 | 38.3 | 86.7 | 43.3 | **100.0** | **63.3** | **100.0** | *50.0* |
| | 50 | *93.3* | 0.0 | 83.3 | 1.7 | 76.7 | 1.7 | 68.3 | 0.0 | 83.3 | 1.7 | *98.3* | **8.3** | **100.0** | **8.3** |
| Backbone | 20 | **100.0** | 63.3 | **100.0** | 96.7 | **100.0** | 96.7 | **100.0** | **100.0** | **100.0** | *98.3* | **100.0** | 96.7 | **100.0** | *96.7* |
| | 50 | *98.3* | 71.7 | **98.3** | 83.3 | 98.3 | *85.0* | 93.3 | 80.0 | 98.3 | 76.7 | **100.0** | 76.7 | **100.0** | *98.3* |
| | 75 | 96.7 | 51.7 | 90.0 | 36.7 | 96.7 | 58.3 | *73.3* | 38.3 | 96.7 | 65.0 | **100.0** | 75.0 | 98.3 | **88.3** |
| | 100 | **100.0** | 50.0 | 83.3 | 15.0 | 81.7 | 33.3 | **53.3** | 11.7 | 93.3 | 50.0 | 98.3 | 73.3 | **100.0** | 83.3 |
| MonoBridge | 10 | **100.0** | 55.0 | **100.0** | 98.3 | **100.0** | 98.3 | **100.0** | 98.3 | **100.0** | 90.0 | **100.0** | **100.0** | **100.0** | **100.0** |
| | 20 | **100.0** | 36.7 | **100.0** | 96.7 | **100.0** | 93.3 | **100.0** | 96.7 | 98.3 | 86.7 | **100.0** | **100.0** | **100.0** | **100.0** |
| | 50 | 98.3 | 13.3 | 96.7 | 66.7 | 93.3 | 71.7 | 68.3 | 26.7 | 96.7 | 58.3 | **100.0** | 61.7 | **100.0** | 83.3 |
| | 75 | **100.0** | 13.3 | 95.0 | 18.3 | 75.0 | 35.0 | 51.7 | 11.7 | 68.3 | 28.3 | 93.3 | *60.0* | **100.0** | **68.3** |
| | 100 | **100.0** | 1.7 | 88.3 | 1.7 | 80.0 | 20.0 | 48.3 | 0.0 | 83.3 | 13.3 | 91.7 | *23.3* | **100.0** | *53.3* |
| | 150 | 96.7 | 3.3 | 80.0 | 1.7 | 83.3 | 1.7 | 58.3 | 0.0 | 78.3 | 1.7 | 93.3 | 21.7 | 90.0 | **31.7** |
| | 200 | 98.3 | 0.0 | 66.7 | 0.0 | 78.3 | 0.0 | 65.0 | 0.0 | 83.3 | 0.0 | 93.3 | *16.7* | 73.3 | 23.3 |

*Table 1.* Satisfiability decision accuracy (Sat., %) and witness validity (Wit., %, SAT only). Each cell averages 10 formulas × 6 verbalizations. Difficulty at 0.5. Best in bold, second in italics.

**Validation bias and variable assignment.** Models can obtain deceptively high decision accuracy by over-predicting SAT. To separate decision priors from constructive reasoning, we additionally require a verifiable SAT witness: when predicting SAT, the model must output an assignment that satisfies all clauses under deterministic checking. We therefore report (i) SAT/UNSAT decision accuracy (Sat.), and (ii) witness validity (Wit.) on SAT instances. To quantify decision bias, we also report SAT prediction rate on UNSAT instances (Appendix Table 5). Each setting contains 10 independently generated formulas; the 6 verbalizations per formula are treated as repeated measurements and aggregated per formula for significance testing. The full per-model SAT/UNSAT confusion matrix is reported in Section M.

**Generator difficulty at comparable formula size.** With $|\mathcal{C}|$ and difficulty fixed (0.5), ImplicationCycle is consistently harder than the SAT generators: UNSAT requires finding a contradiction certificate (reachability $x \rightsquigarrow \neg x$ and $\neg x \rightsquigarrow x$) that depends on long-range implication integration, whereas EquivalenceCore and Backbone often allow local propagation over repeated equivalence patterns. Llama-3.3-70B-Instruct also shows a strong SAT bias, predicting SAT on 88.3% of UNSAT instances (Appendix Table 5). At $|\mathcal{C}| = 50$, mean accuracy suggests ImplicationCycle and EquivalenceCore are hardest, then Backbone, with MonoBridge easiest.

## 5.1. Impact of Difficulty Parameters

To assess whether generator parameters matter beyond $|\mathcal{C}|$, we sweep each parameter over a 7-point grid

$0, 0.2, 1/3, 0.5, 2/3, 0.8, 1$ while fixing the others at $0.5$. Each formula is one sample (10 per setting), with verbalizations aggregated per formula. We apply Friedman's test across levels and Benjamini–Hochberg correction within each ablation family. Sweeps are run at clause counts near each generator's transition region: $|\mathcal{C}| = 75$ for ImplicationCycle, MonoBridge, and Backbone, and $|\mathcal{C}| = 20$ for EquivalenceCore. Appendix Table 6 shows that several effects are significant only under the 7-point sweep or only after aggregation across sizes. We therefore interpret these trends as targeted brittleness regimes rather than universal monotonic effects.

**MonoBridge parameters.** The positive-part ratio has a significant effect ($p < 0.05$ at $|\mathcal{C}| = 75$): instances are hardest near a balanced split (ratio $\approx 0.5$) and easier when one monotone region dominates, consistent with a simple majority-assignment heuristic in the imbalanced regime and the need to reason about the cross-polarity bridge when balanced. Variable density also has a strong effect ($p \ll 0.001$): low density favors simplification via repeated variables, intermediate density produces a performance valley, and some large models recover at high density as the bridge becomes more salient. Qwen3-Next-80B and GPT-OSS-120B show improved performance around density 0.6 to 0.8, which is consistent with the bridge becoming more salient. The bridge position is not significant under our current protocol. One possible explanation is that many models do not process clauses strictly sequentially: they often restate the problem as an unordered set of constraints before attempting simplification.

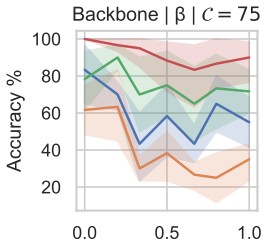
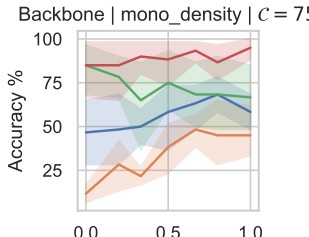
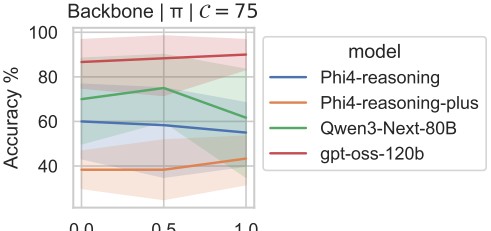

*Figure 1.* Main three difficulty parameters for Backbone at $|\mathcal{C}| = 75$. Shade corresponds to 95% confidence interval. For $\pi$, $\pi = 0$ for a negative and $\pi = 1$ for a positive monotone part, while at $\pi = 0.5$ the sign is randomly chosen.

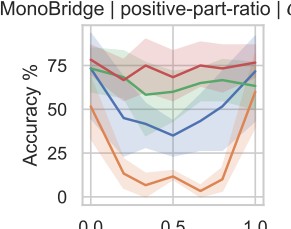
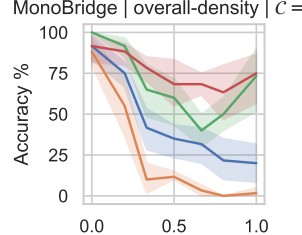
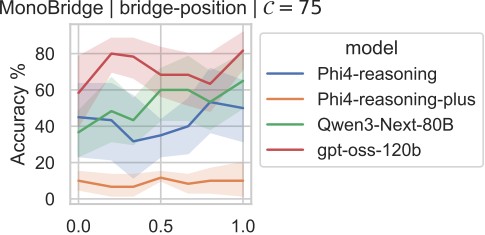

*Figure 2.* Main three difficulty parameters for MonoBridge at $|\mathcal{C}| = 75$. Shade corresponds to 95% confidence interval.

**Backbone parameters.** The three difficulty parameters identified for Backbone have a limited impact on the performance compared to, for instance, the number of clauses. First, the *backbone fraction* $\beta$ has the most significant effect on the performance ($p$-value $< 0.005$ for $|\mathcal{C}| = 75$ and $p$-value $< 0.02$ overall), but this effect is marginal compared to the effect of the difficulty parameters of the other generators, with a drop of around 10% in the average performance between $\beta = 0$ and $\beta = 1$. This indicates that the backbone part, which has pairs of related clauses, is more complex to handle than the monotone part. While the *density of the monotone part* (mono density) has the next most significant impact on performance, it is significant only when considering $|\mathcal{C}| = 75$ where we observe an increase in performance of up to 20% for the worst model. Higher monotone density means fewer distinct literals and stronger propagation. Finally, the polarity $\pi$ of the monotone part has no significant impact on the performance, with models manipulating literal polarity without issues in the CoT traces we analyzed (all $p$-values $> 0.9$).

**EquivalenceCore parameter.** The free-to-bound variable ratio has a strong and consistent effect (Figure 3), with significant effects both overall and in the 7-point sweep at $|\mathcal{C}| = 20$. ($p$-value $< 0.005$ for $|\mathcal{C}| = 20$ and $p$-value $= 0.31$ overall). High values of the ratio, corresponding to low propagation, are significantly harder to solve than low ratios, as the models rely on early simplification of equivalence classes through propagation. Across CoT traces, all models managed to properly identify equivalences from the 2–CNF in its construction order.

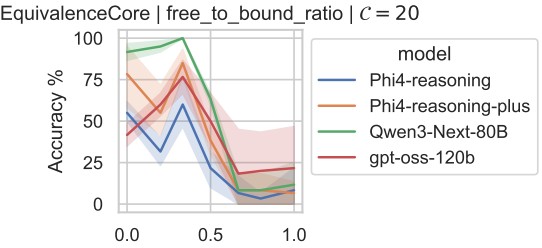

*Figure 3.* Main difficulty parameter for EquivalenceCore at $|\mathcal{C}| = 20$. Shade corresponds to 95% confidence interval.

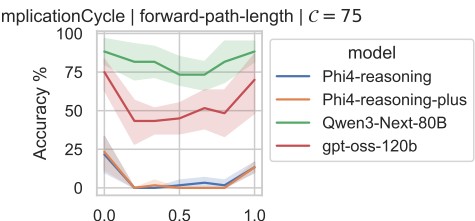

*Figure 4.* Main difficulty parameter for implication cycle at $|\mathcal{C}| = 75$. Shade corresponds to 95% confidence interval.

**ImplicationCycle parameter.** The ImplicationCycle split parameter ($k/m$) affects performance mainly at larger sizes: extreme splits create one short side of the contradiction cycle, offering a shorter certificate path to track, while near-balanced splits force long-range reachability reasoning in both directions. The effect is significant at $|\mathcal{C}| = 75$ but does not consistently generalize across sizes, so we see it as a targeted brittleness regime rather than a universal trend.

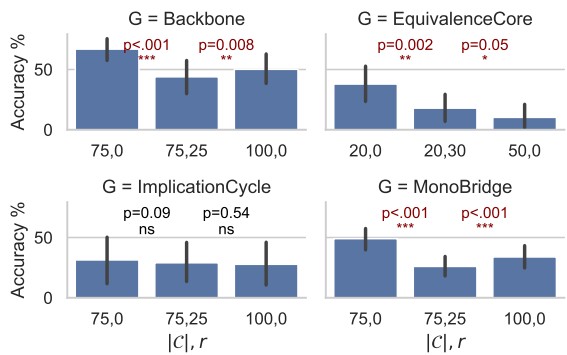

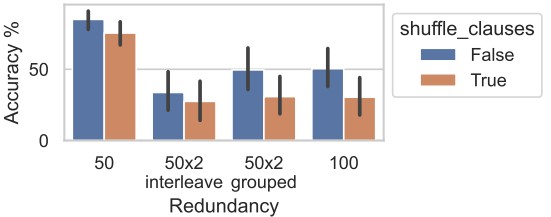

Figure 7. Comparison of accuracy when duplicating the clauses. All difficulty parameters are set at $0.5$. Error bars are 95% confidence intervals. For UNSAT, accuracy is measured on the prediction of satisfiability, and on variable assignment for SAT problems.

Figure 5. Effect of adding non-core filler clauses. We compare fixed-core settings with and without filler, and matched-total-size settings where a smaller core plus filler is compared to a larger core alone. Error bars are 95% confidence intervals across models and verbalization templates.

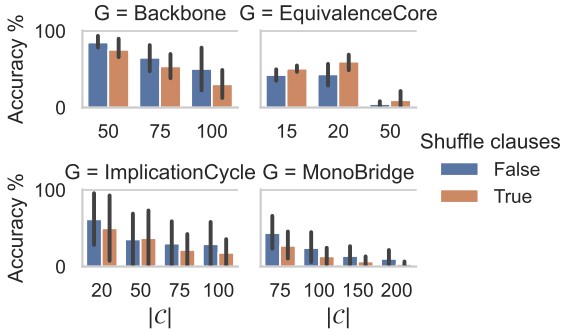

Figure 6. Comparison of accuracy between the construction order and a shuffling of the clauses. All difficulty parameters are set at $0.5$. Error bars are 95% confidence intervals. For UNSAT problems, accuracy is measured on the prediction of satisfiability, and on variable assignment for SAT problems.

## 5.2. Impact of Non-core Variables

To test sensitivity to variables irrelevant to satisfiability, we add *filler* clauses over fresh variables. Let $V_{\text{fresh}}$ be disjoint from the core variable set and define literals $\ell_i \in v_i, \neg v_i$ for $v_i \in V_{\text{fresh}}$. We sample filler clauses from pairs of these literals, choosing $|V_{\text{fresh}}|$ so filler and core clauses have comparable variable density. We use two comparison regimes: fixed core size and matched total clause count. Concretely, we compare $(|C| = 75, r = 25)$ to $(75, 0)$ and $(100, 0)$, and $(|C| = 20, r = 30)$ to $(20, 0)$ and $(50, 0)$. At difficulty $0.5$, trends are visible but rarely significant, so we report results on the 3-level grid from Section 5.1 $(0.2, 0.5, 0.8)$. For all generators except ImplicationCycle, adjacent settings differ significantly (paired $t$-tests; see Figure 5). Filler clauses significantly hurt EquivalenceCore, Backbone, and MonoBridge when the core size is fixed. Under matched total clauses, EquivalenceCore behaves dif-

ferently: a smaller equivalence core plus filler can be easier than a larger core alone. Trace inspection suggests models prioritize equivalences, but when they fail to compress equivalence classes, they repeatedly re-verify local consistency, making equivalences more costly than filler. For Backbone and MonoBridge, filler mainly adds noise, since their monotone regions already resemble filler structure, which further obscures the global pattern.

## 5.3. Sensitivity to Clause Order

We compare in Figure 6 our four main models under two clause orders: *(i)* the generator construction order (see Section 3) and *(ii)* a random permutation. Clause order significantly affects performance for all generators (paired $t$-tests, $p < 0.003$), with a weaker but still significant effect for ImplicationCycle ($p = 0.035$). For EquivalenceCore, shuffling is beneficial. As discussed in Section 5.2, when implication structure is poorly formalized, contiguous implication chains can act as misleading local patterns; shuffling breaks these groupings by separating related clauses, reducing early commitment to an incorrect structure.

## 5.4. Ability to Handle Repeated Patterns

Figure 7 shows results for Backbone at medium difficulty (all parameters set to $0.5$) under clause repetition. We duplicate a base set of 50 clauses to obtain 100 clauses, using a fresh variable-to-entity mapping for the copy. We test two layouts: *grouped*, where the copy forms the last 50 clauses, and *interleave*, where copied clauses are randomly interleaved while preserving their internal order. We also compare against random clause permutations. Performance at 50 clauses differs significantly from all other variants (independent $t$-tests, $p < 0.001$). Shuffling significantly degrades performance within each setting (paired $t$-tests, $p < 0.001$), except for 50×2 *interleave* ($p = 0.036$). Differences between the 50×2 and 100-distinct-clauses conditions are not significant ($p > 0.1$), suggesting that repeating clauses is about as hard as adding distinct ones. Reasoning traces indicate that models rarely detect or exploit repeated patterns.

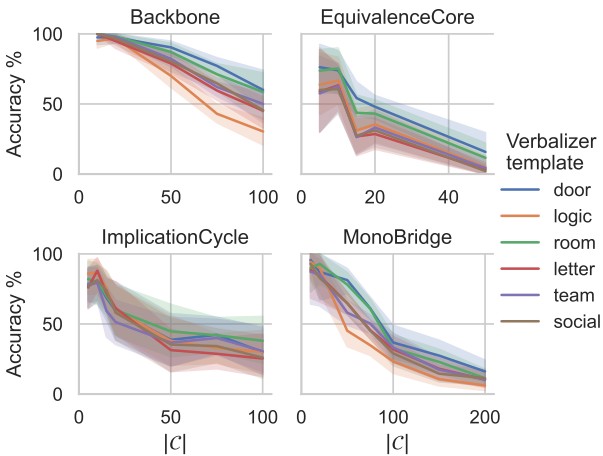

*Figure 8.* Decision accuracy on UNSAT and SAT witness validity against the number of clauses for each generator, depending on the verbalization template used. Shade is the 95% confidence interval.

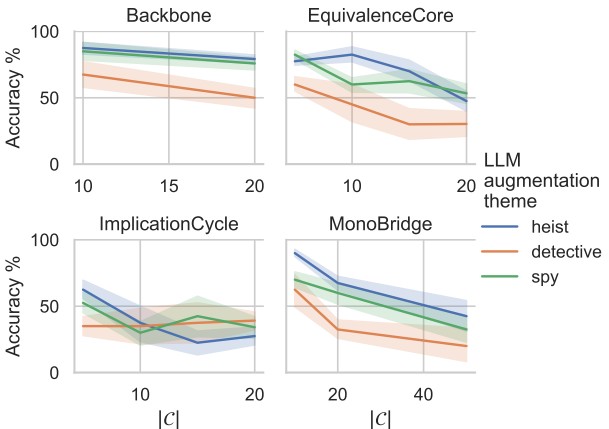

*Figure 9.* Decision accuracy on UNSAT and SAT witness validity against number of clauses for each generator, depending on the used LLM verbalization. Error bars are 95% confidence intervals.

### 5.5. Verbalization strategy

As shown in Figure 8, template-only verbalizations yield similar performance across templates, indicating little sensitivity to surface wording when the clause structure is explicit. In contrast, LLM verbalizer substantially degrades accuracy (Figure 9): at comparable clause sizes, performance drops by about 25 points relative to Figure 8. We attribute this mainly to increased contextual noise and weaker structural cues, which make clause extraction harder even though the underlying 2-CNF is information-preserving and can be recovered when models are explicitly prompted to do so. Across themes, detective narratives are hardest, followed by spy, then heist, consistent with the intended stylistic bias. Detective stories tend to be more indirect, which further obscures the logical form. For ImplicationCycle, theme effects are smaller: the shared-literal chain induces more co-

herent narratives, improving verbalization consistency and partially mitigating theme differences. Overall, templates make the 2-CNF structure easy to recognize, whereas LLM narratives often impede structure recovery and lead models to miss global logical patterns. Section L reports mean prompt-token counts for each verbalizer. At a fixed $|\mathcal{C}|$, the LLM verbalizer produces prompts roughly 4–10× longer than the templates (Section L).

## 6. Conclusion

We introduced a diagnostic 2-SAT benchmark based on parameterized 2-CNF families with controllable implication-graph structure, together with semantics-preserving perturbations such as clause reordering, variable renaming, and non-core fillers. The benchmark is designed to test whether solver-like behavior remains stable when satisfiability is unchanged but presentation, distractor load, or witness structure is varied. Across several reasoning models, these targeted interventions induce sharp performance shifts even at fixed instance size. Long-cycle UNSAT instances are particularly brittle, and the gap between decision accuracy and assignment validity shows that correct SAT/UNSAT predictions can mask weak witness construction. Sensitivity to ordering, fillers, and repetition further indicates limited invariance and limited reuse of intermediate structure. The restriction to 2-SAT is deliberate. Its implication graph and SCC decomposition provide a fine-grained and interpretable mapping from graph-theoretic properties, such as cycle length, backbone fraction, and bridge placement, to controlled diagnostic probes. Extending our work to richer constraint classes, while preserving this separation between structure and search hardness, is a natural direction for future work.

## Acknowledgments

This work was supported by ANR-22-CE23-0002 ERIANA and ANR-25-CE23-7498 DesiRes, and by access to the HPC resources of IDRIS under the allocation 2026-AD011013338 made by GENCI.

## Impact Statement

This paper aims to advance the understanding of reasoning capabilities and failure modes in LRMs through controlled diagnostic evaluation on logical satisfiability problems. We do not foresee direct negative societal consequences from this work. By identifying brittleness regimes that are not captured by aggregate accuracy metrics, the proposed benchmark may support more careful evaluation of LLM-based reasoners, especially in applications where logical consistency and verifiable outputs are important.

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

# A. Background

Let $\mathcal{V}$ be a finite set of variables.

A *literal* is either a variable $p$ (positive) or its negation $\neg p$ (negative). Two literals are *complementary* if one is the negation of the other.

A *clause* is a disjunction of literals. A formula is in *Conjunctive Normal Form* (CNF) if it is a conjunction of clauses. A 2–CNF formula is a CNF formula where each clause contains at most two literals.

An *interpretation* is a function $\omega : \mathcal{V} \longrightarrow \{0,1\}$, where 1 stands for *true* and 0 for *false*. It is extended to formulas as follows: $\omega(\neg\varphi) = 1 - \omega(\varphi)$, $\omega(\varphi \wedge \psi) = \min(\omega(\varphi), \omega(\psi))$, and $\omega(\varphi \vee \psi) = \max(\omega(\varphi), \omega(\psi))$.

A model of a formula $\varphi$ is an interpretation $\omega$ that satisfies the formula: $\omega(\varphi) = 1$. In particular, $\omega$ is a model of a CNF formula $\varphi$ if and only if for each clause $c$ in $\varphi$, there exists a literal $l$ in $c$ such that $\omega(l) = 1$.

A formula is *satisfiable* if and only if it has at least one model; otherwise, it is *unsatisfiable*.

**Definition A.1** (Implication Graph). Let $\varphi$ be a 2–CNF formula. The *implication graph* of $\varphi$ is a directed graph $G = (V, E)$, where $V = Lit(Var(\varphi))$ and $E = \bigcup \{\{(\bar{l}, l'), (\overline{l'}, l)\} : l \vee l' \in \varphi \text{ or } (l \in \phi \text{ and } l' = l)\}$.

We use $l \rightsquigarrow l'$ to denote that there is a path from $l$ to $l'$ in the implication graph.

**Definition A.2** (Proof Cycle). Let $\varphi$ be an unsatisfiable 2–CNF formula. A *proof cycle* of the unsatisfiability of $\varphi$ is a cycle $\boxed{l} \rightarrow \boxed{l_1} \rightarrow \cdots \boxed{l_k} \rightarrow \boxed{l}$ in the implication graph of $\phi$, where there exists $i \in \{1, \ldots, k\}$ such that $l_i = \bar{l}$.

**Theorem A.3.** *A 2–CNF formula is unsatisfiable iff it admits a proof cycle.*

# B. UNSAT Generator using Implication Cycles

We create one clause per implication shown above:

$$\boxed{\begin{aligned} &(\neg\ell_1 \vee \ell_2), (\neg\ell_2 \vee \ell_3), \ldots, (\neg\ell_{k-1} \vee \ell_k), (\neg\ell_k \vee \neg\ell_1), \\ &(\neg\neg\ell_1 \vee \ell_{k+1}), (\neg\ell_{k+1} \vee \ell_{k+2}), \ldots, (\neg\ell_{m-1} \vee \ell_m), (\neg\ell_m \vee \ell_1). \end{aligned}}$$

We describe our generator in Algorithm 1.

*Example* 1 (UNSAT core with $m{=}5$, $k{=}3$). Let $(\ell_1, \ldots, \ell_5) = (v_1, \neg v_2, v_3, v_4, \neg v_5)$. Core clauses:

$$(\neg v_1 \vee \neg v_2), (v_2 \vee v_3), (\neg v_3 \vee \neg v_1),$$
$$(v_1 \vee v_4), (\neg v_4 \vee \neg v_5), (v_5 \vee v_1).$$

This forces $v_1 \rightsquigarrow \neg v_1$ and $\neg v_1 \rightsquigarrow v_1$, hence UNSAT.

# C. SAT Generator via Free Variables

For each $y \in \mathcal{V} \setminus F$:

$$\tau(y) = p(y): \quad (\neg y \vee p(y)) \wedge (\neg p(y) \vee y), \qquad \tau(y) = \neg p(y): \quad (\neg y \vee \neg p(y)) \wedge (p(y) \vee y).$$

This generator is described in Algorithm 2.

*Example* 2. Let $F = \{p\}$ (free) and define $q \leftrightarrow \neg p$, $r \leftrightarrow p$. Clauses:

$$(\neg q \vee \neg p) \wedge (p \vee q) \wedge (\neg r \vee p) \wedge (\neg p \vee r).$$

There are exactly two models, extending $p \in \{0, 1\}$.

# D. Backbone-based Method

**Pseudocode:**

---

**Algorithm 1** GENERATEUNSAT2CNF($n, m, k, r, shuffle$)

---

**Require:** $n \geq m \geq 2, 1 \leq k \leq m, r \geq 0$, $shuffle \in \{\text{true}, \text{false}\}$
 1: Create variables $\mathcal{V} = \{x_1, \ldots, x_n\}$; pick distinct $v_1, \ldots, v_m \in \mathcal{V}$
 2: For $i = 1..m$, choose a polarity $\ell_i \in \{v_i, \neg v_i\}$
 3: Initialize clause set $\mathcal{C} \leftarrow \emptyset$ {Forward part}
 4: **for** $i = 1$ to $k - 1$ **do**
 5: $\quad \mathcal{C} \leftarrow \mathcal{C} \cup \{(\neg \ell_i \vee \ell_{i+1})\}$
 6: **end for**
 7: $\mathcal{C} \leftarrow \mathcal{C} \cup \{(\neg \ell_k \vee \neg \ell_1)\}$ {Backward part}
 8: $\mathcal{C} \leftarrow \mathcal{C} \cup \{(\neg \neg \ell_1 \vee \ell_{k+1})\}$
 9: **for** $i = k + 1$ to $m - 1$ **do**
10: $\quad \mathcal{C} \leftarrow \mathcal{C} \cup \{(\neg \ell_i \vee \ell_{i+1})\}$
11: **end for**
12: $\mathcal{C} \leftarrow \mathcal{C} \cup \{(\neg \ell_m \vee \ell_1)\}$ {Fillers}
13: **for** $j = 1$ to $r$ **do**
14: $\quad$ Add a filler clause $(\lambda \vee \lambda')$ over $\mathcal{V}$ (optionally avoid tautologies)
15: **end for**
16: **if** $shuffle$ **then**
17: $\quad$ Randomly permute $\mathcal{C}$
18: **end if**
19: Output $\Phi = \bigwedge_{C \in \mathcal{C}} C = 0$

---

**Algorithm 2** GENERATESAT2CNF($n, f$)

---

**Require:** $n \geq 1, 0 \leq f \leq n$
 1: Create variables $\mathcal{V} = \{x_1, \ldots, x_n\}$; choose free set $F \subseteq \mathcal{V}$ with $|F| = f$
 2: $\mathcal{C} \leftarrow \emptyset$
 3: **for** each $y \in \mathcal{V} \setminus F$ **do**
 4: $\quad$ Pick $p(y) \in F$ and sign $s \in \{+1, -1\}$
 5: $\quad$ **if** $s = +1$ **then**
 6: $\quad\quad \mathcal{C} \leftarrow \mathcal{C} \cup \{(\neg y \vee p(y)), (\neg p(y) \vee y)\}$
 7: $\quad$ **else**
 8: $\quad\quad \mathcal{C} \leftarrow \mathcal{C} \cup \{(\neg y \vee \neg p(y)), (p(y) \vee y)\}$
 9: $\quad$ **end if**
10: **end for**
11: Output $\Phi = \bigwedge_{C \in \mathcal{C}} C = 0$

---

1. Input: $n$, $B \subseteq \mathcal{V}$, planted $b$, orientation $\pi \in \{+, -\}$, $m_{\text{mono}}$.

2. $F \leftarrow \mathcal{V} \setminus B$; initialize clause multiset $\mathcal{C} \leftarrow \emptyset$.

3. For each $x \in B$: introduce $a_x$ and add $(x \vee a_x), (x \vee \neg a_x)$ if $b(x) = 1$, else $(\neg x \vee a_x), (\neg x \vee \neg a_x)$.

4. Repeat $m_{\text{mono}}$ times: sample $y, z \in F$, $y \neq z$, and add $(\neg y \vee \neg z)$ if $\pi = -$, else $(y \vee z)$.

5. Output $\Phi^\star = \bigwedge_{C \in \mathcal{C}} C$.

# E. Monotone Split with a Moving Cross-Polarity Bridge

**Construction:**

1. **Disjoint variable sets.** Partition the variables into $P = \{p_1, \ldots, p_{n_+}\}$ and $N = \{n_1, \ldots, n_{n_-}\}$ with $P \cap N = \emptyset$.

2. **Positive-monotone core $\Phi_+$.** Generate $m_+$ clauses, each $(p_i \vee p_j)$ for some $p_i, p_j \in P$ (avoid tautologies and duplicates if desired).

3. **Negative-monotone core $\Phi_-$.** Generate $m_-$ clauses, each $(\neg n_i \vee \neg n_j)$ for some $n_i, n_j \in N$.

4. **Bridge clause $B$.** Pick endpoints $p^\star \in P$ and $n^\star \in N$ and set

$$B := (\neg p^\star \vee n^\star).$$

5. **Ordering (moving the bridge).** Fix an overall ordering of $\Phi_+ \cup \Phi_-$ (e.g., all $\Phi_+$ then all $\Phi_-$ or any shuffle). Let $M = m_+ + m_-$ be the number of monotone clauses. Choose an index $s \in \{1, \ldots, M+1\}$ and *insert $B$ as the $s$-th clause*. Varying $s$ across instances moves where the solver first encounters the coupling.

**Proposition E.1** (Satisfiable by construction). *The formula $\Phi := \Phi_+ \wedge \Phi_- \wedge B$ is satisfiable.*

For example, the assignment that set all $P$ to 1 and set $n^\star$ to 1 and all other $N \setminus \{n^\star\}$ to 0 satisfies $\Phi_+$ (positives are 1), satisfies $B$ (since $n^\star = 1$), and satisfies every $(\neg n_i \vee \neg n_j)$ in $\Phi_-$ (at least one of $n_i, n_j$ is 0).

**Backtracking trigger.** If a solver's early decisions tend to make $p^\star = 1$ (natural for $\Phi_+$) and $n^\star = 0$ (natural for $\Phi_-$), then when $B$ is finally processed it becomes $(\neg 1 \vee 0)$ and conflicts, forcing a revision (unit propagation or backtrack). By changing the insertion index $s$, we control when this conflict becomes visible.

**Pseudocode**

1. Input: $n_+, n_-, m_+, m_-, p^\star \in P, n^\star \in N$, insertion index $s \in [1, M+1]$.

2. Build lists $C_+$ of $m_+$ clauses $(p \vee p')$ over $P$ and $C_-$ of $m_-$ clauses $(\neg n \vee \neg n')$ over $N$.

3. Concatenate $C := \text{order}(C_+, C_-)$ (any fixed order or shuffle); let $M = |C|$.

4. Insert $B = (\neg p^\star \vee n^\star)$ at position $s$ in $C$.

5. Output $\Phi = \bigwedge_{i=1}^{M+1} C[i]$.

# F. Symmetry/Redundancy Probe

**Construction**

1. **Base formula.** Let $\Psi = \bigwedge_{c \in \mathcal{C}} c$ be any 2–CNF over variables $V = \{x_1, \ldots, x_n\}$.

2. **Disjoint renaming.** Choose a bijection $\rho : V \to V' = \{x'_1, \ldots, x'_n\}$ with $V \cap V' = \varnothing$. Extend $\rho$ to literals and clauses in the obvious way (e.g., $\rho(\neg x) = \neg \rho(x)$, $\rho(\ell_1 \vee \ell_2) = \rho(\ell_1) \vee \rho(\ell_2)$).

3. **Concatenation.** Define
$$\Phi := \Psi \wedge \rho(\Psi).$$
Optionally, order the clauses as (i) all of $\Psi$ then all of $\rho(\Psi)$ (grouped), or (ii) an interleaving of the two clause lists (to blur copy boundaries).

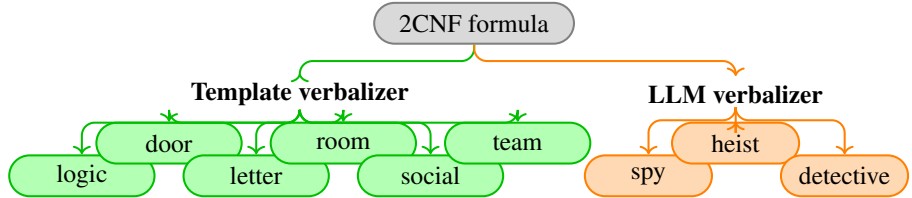

*Figure 10.* Summary of the verbalization process

# G. Verbalization Templates and Prompts

## G.1. Template Verbalizers

Each template verbalizer converts a 2-CNF clause $(\ell_1 \vee \ell_2)$ into a natural language sentence following a fixed pattern. The verbalizer also provides a context preamble and a final yes/no question. Table 2 summarizes the six schemes.

| Scheme | True value | False value | Entity naming | Context preamble |
|---|---|---|---|---|
| logic | true | false | Letters: A, B, ..., Z, AA, AB, ... | Consider these logical relationships: |
| letter | has | doesn't have | Person names | Consider these rules about who has the letter: |
| team | Red | Blue | Person names | We have team assignment rules: |
| social | attends | doesn't attend | Person names | You're planning a party with these attendance conditions: |
| room | lit | dark | room 1, room 2, ... | A building has lighting rules: |
| door | open | closed | door 1, door 2, ... | A facility has door coupling rules: |

*Table 2.* Summary of the six template verbalizers. Each row shows the scheme name, the context preamble, the true/false value labels, and the entity naming convention.

**Clause-to-text patterns.** Below are the exact sentence templates used for each scheme. In all cases, $A$ and $B$ denote the entity names of the two literals' variables, and the polarity determines which value (true or false) is used.

- **logic:** `Either {A} is true or {B} is false (or both).`

- **letter:** `Either {A} has the letter or {B} doesn't have the letter (or both).`

- **team:** `Either {A} is on the Red team or {B} is on the Blue team.`

- **social:** `{A} attends or {B} doesn't attend (or both).`

- **room:** `room {i} is lit or room {j} is dark (or both).`

- **door:** `door {i} is open or door {j} is closed (or both).`

**Final questions.**

- **logic:** Can you assign truth values to all statements without creating a contradiction?

- **letter:** Is there a consistent way for people to hold or not hold the letter?

- **team:** Is there a valid team assignment that satisfies all constraints?

- **social:** Can you create a guest list that respects everyone's conditions?

- **room:** Can all the lighting rules be satisfied simultaneously?

- **door:** Is there a configuration of doors that satisfies all the rules?

**Assembled output.**   For each formula, the verbalizer concatenates the context preamble, one sentence per clause, a blank line, and the final question. For example, with the `logic` scheme and clauses $(\neg A \lor B)$, $(A \lor \neg C)$:

```
Consider these logical relationships:
Either A is false or B is true (or both).
Either A is true or C is false (or both).

Can you assign truth values to all statements
without creating a contradiction?
```

### G.2. Evaluation Prompt

The evaluation prompt is appended after the verbalized formula. Two variants exist depending on whether template or LLM verbalization is used.

**Template verbalizer prompt.**

```
{verbalized formula text}

Think step by step.

If a valid assignment exists, provide it as JSON
using "{true_val}" or "{false_val}" as values.
Example format:
{
  "{entity_1}": "{true_val}",
  "{entity_2}": "{false_val}",
  "{entity_3}": "{true_val}"
}

End your response with exactly one of:
- "The answer is: Yes" (if a valid assignment exists)
- "The answer is: No" (if no valid assignment exists)
```

where {`true_val`} and {`false_val`} are the scheme-specific values from Table 2, and the example JSON shows the first three entities.

**LLM verbalizer prompt.**

```
{augmented story text}

Think step by step.

If a valid assignment exists, provide it as JSON
with one of the allowed values for each entity.

Entities and their possible values:
  - "{entity_name}": "{true_desc}" or "{false_desc}"
  ...

Example JSON format:
{
  "entity_name": "value"
}

End your response with exactly one of:
- "The answer is: Yes" (if a valid assignment exists)
- "The answer is: No" (if no valid assignment exists)
```

where each entity is listed with its theme-specific true/false descriptions.

### G.3. LLM Story Generation Prompts

The LLM verbalizer uses a multi-turn conversation to generate narrative clues. Below are the exact prompts used at each step.

**Step 1: Global context generation.**

```
You are a mystery story writer. Create a short
introduction for a detective story.

CHARACTERS/OBJECTS in this story:
- {entity_1_name}
- {entity_2_name}
...

Requirements:
- Set the scene for a mystery with clues to discover
- Briefly introduce each character/object naturally
  in the narrative
- Do NOT mention any specific rules or relationships
  yet
- Keep it brief and atmospheric

Write only the introduction paragraph.
```

**Step 2: Per-clause narrative generation.**   For each clause $(\ell_1 \vee \ell_2)$, the following prompt is appended to the conversation:

```
Write clue {i}/{total} for our mystery story.

This clue must express that AT LEAST ONE of these
statements is true:
- Statement A: {entity_1_name} {state_1}
- Statement B: {entity_2_name} {state_2}

(It's possible both are true, but at least one
MUST be true.)

Write 2-3 sentences that naturally convey this
logical rule through detective reasoning, witness
testimony, or evidence analysis. Be creative but
ensure the logic is clear.

Write ONLY the clue paragraph:
```

**Step 3a: Entity extraction (validation).**

```
From this clue, identify the TWO main entities
being discussed.

CLUE: "{generated_paragraph}"

AVAILABLE ENTITIES:
["entity_1", "entity_2", ...]

Which two entities does this clue primarily discuss?
Respond with ONLY a JSON array of exactly 2 entity
names: ["entity1", "entity2"]
```

| Model | Model ID | Param. | Max Tokens | License |
|---|---|---|---|---|
| Phi 4 reasoning | microsoft/Phi-4-reasoning | 15B | 32K | MIT |
| Phi 4 reasoning plus | microsoft/Phi-4-reasoning-plus | 15B | 32K | MIT |
| Qwen3-Next 80B | Qwen/Qwen3-Next-80B-A3B-Thinking | 81B | 262K | Apache 2.0 |
| GPT-OSS-120B | openai/gpt-oss-120b | 120B | 131K | Apache 2.0 |
| OLMo-3 32B | allenai/Olmo-3.1-32B-Think | 32B | 66K | Apache 2.0 |
| QwQ 32B | Qwen/QwQ-32B | 33B | 41K | Apache 2.0 |
| Llama-3.3 70B-Instruct | meta-llama/Llama-3.3-70B-Instruct | 71B | 66K | Llama 3.3 |

*Table 3.* Overview of the reasoning models.

| Model | Max Tokens | Truncation Rate (%) |
|---|---|---|
| Phi-4-reasoning | 16 384 | 14.9 |
| Phi-4-reasoning-plus | 16 384 | 30.5 |
| Qwen3-Next 80B | 32 768 | 1.3 |
| GPT-OSS-120B | 32 768 | 0.0 |
| OLMo-3 32B | 32 768 | 3.0 |
| QwQ 32B | 32 768 | 3.6 |
| Llama-3.3 70B-Instruct | 32 768 | 0.0 |

*Table 4.* Truncation rates by model under fixed output budget.

**Step 3b: State extraction (validation).** For each of the two extracted entities:

```
In this clue, what is stated about {entity_name}?

CLUE: "{generated_paragraph}"

The clue says that {entity_name}:
A) {true_description}
B) {false_description}

Which state does the clue indicate?
Answer with ONLY "A" or "B".
```

If validation fails after 3 attempts, a programmatic fallback sentence is used:

```
Either {entity_1} {state_1}, or {entity_2} {state_2} (or both).
```

# H. Model Information

Table 3 reports general model information.

# I. Truncation Rate

Table 4 reports output truncation rates per model.

# J. SAT Prediction Bias on UNSAT Instances

Table 5 reports the proportion of UNSAT instances for which each model incorrectly predicted SAT. A high rate indicates a tendency to over-predict satisfiability regardless of the actual logical structure. Llama-3.3-70B-Instruct exhibits a particularly strong SAT bias (88.3%), suggesting it defaults to predicting satisfiability rather than performing genuine constraint reasoning.

| Model | SAT Prediction (%) |
|---|---|
| Phi-4-reasoning | 33.3 |
| Phi-4-reasoning-plus | 20.2 |
| Qwen3-Next 80B | 10.5 |
| GPT-OSS-120B | 26.4 |
| OLMo-3 32B | 21.7 |
| QwQ 32B | 34.0 |
| Llama-3.3 70B-Instruct | 88.3 |

*Table 5.* SAT prediction rate on UNSAT instances (ImplicationCycle generator). Higher values indicate stronger bias toward predicting satisfiability.

## K. Statistical Significance Results

The significance results for Friedman's test for repeated measurements across difficulty parameter levels with Benjamini–Hochberg correction are reported in Table 6.

## L. Estimated Mean Token Count per Verbalizer

Table 7 reports the estimated mean token count (computed as $\lfloor \text{chars}/4 \rfloor$ on the rendered prompts) as a function of the number of clauses $|\mathcal{C}|$, for the six template verbalizers used in the paper. Token count grows nearly linearly with $|\mathcal{C}|$ in every verbalizer; the `letter` verbalizer is consistently the longest (about $1.5\times$ the most compact), while `logic` and `room` are the shortest. The spread between verbalizers at fixed $|\mathcal{C}|$ is bounded by a factor of $\sim 1.6$, while the gap between *template* and *LLM-generated* verbalizations (multi-sentence paragraphs per clause) is substantially larger and explains part of the drop reported in Section 5.5.

### Template vs. LLM-generated verbalization

The LLM verbalizer (see Section G) renders each clause as a multi-sentence paragraph, which yields substantially longer prompts than the template verbalizers at matched $|\mathcal{C}|$. Table 8 reports the actual mean prompt-token count measured on the GPT-OSS-120B run, comparing the matched (generator, $|\mathcal{C}|$) pairs available in both normal and augmented configurations. This systematic increase in prompt length partly explains the accuracy drop reported for the LLM verbalizer in Section 5.5, distinct from the effect of surface wording.

## M. Per-Model Confusion Matrix

Table 9 reports the per-model confusion matrix for the SAT vs. UNSAT decision, aggregated over all generators and parameter settings of the main evaluation (Table 1). Convention: *Positive* = SAT. Llama-3.3-70B-Instruct stands out with a large false-positive count (FP = 371), consistent with the strong SAT-prediction bias already reported in Table 5. Qwen3-Next-80B achieves the highest accuracy (96.3%) and the most balanced error profile.

## N. Proprietary Reasoning Models

We complement the open-weight evaluation of Table 1 with three recent proprietary reasoning models—o3-mini, o4-mini, and GPT-5.4-mini—run on the same prompts and the same protocol (10 formulas × 6 template verbalizers per cell). Table 10 reports per-(generator, $|\mathcal{C}|$) decision accuracy (Sat., %) and witness validity (Wit., %, SAT instances only).

## O. GPT-OSS-120B: Effect of Reasoning Effort

GPT-OSS exposes three reasoning-effort levels (`low`, `medium`, `high`) through the `reasoning_effort` field of its chat template when served locally via vLLM. The results reported in the main paper (Table 1) use the default setting (`medium`). To characterise how the compute budget allocated to internal reasoning interacts with the difficulty axes of our benchmark, we report in Table 11 the Sat. and Wit. at all three effort levels.

| Generator | Difficulty parameter | $|\mathcal{C}|$ | $p$-value |
|---|---|---|---|
| Backbone | monotone density | Overall | $3.363e^{-3}$ |
| | | 20 | $4.724e^{-1}$ |
| | | 50 | $3.813e^{-2}$ |
| | | 75 | $4.724e^{-1}$ |
| | | 75 7-point | $3.125e^{-1}$ |
| | | 100 | $3.877e^{-2}$ |
| | $\beta$ | Overall | $1.865e^{-2}$ |
| | | 20 | $1.462e^{-1}$ |
| | | 50 | $5.971e^{-2}$ |
| | | 75 | $3.877e^{-2}$ |
| | | 75 7-point | $4.310e^{-3}$ |
| | | 100 | $3.679e^{-1}$ |
| | $\pi$ | 75 7-point | $9.355e^{-1}$ |
| EquivalenceCore | Free-to-bound variable ratio | Overall | $2.754e^{-5}$ |
| | | 15 | $1.832e^{-2}$ |
| | | 20 | $1.832e^{-2}$ |
| | | 20 7-point | $1.296e^{-3}$ |
| | | 50 | $8.208e^{-2}$ |
| ImplicationCycle | Forward path relative length | Overall | $8.116e^{-1}$ |
| | | 5 | $7.643e^{-2}$ |
| | | 10 | $5.488e^{-1}$ |
| | | 15 | $3.679e^{-1}$ |
| | | 20 | $2.574e^{-1}$ |
| | | 50 | $4.412e^{-1}$ |
| | | 75 | $2.725e^{-1}$ |
| | | 75 7-point | $7.257e^{-3}$ |
| | | 100 | $5.292e^{-1}$ |
| MonoBridge | Bridge relative position | Overall | $1.869e^{-1}$ |
| | | 50 | $1.889e^{-1}$ |
| | | 75 | $7.788e^{-1}$ |
| | | 75 7-point | $2.154e^{-1}$ |
| | | 100 | $1.738e^{-1}$ |
| | | 150 | $1.496e^{-1}$ |
| | | 200 | $1.353e^{-1}$ |
| | Overall variable density | Overall | $2.073e^{-7}$ |
| | | 50 | $4.979e^{-2}$ |
| | | 75 | $1.832e^{-2}$ |
| | | 75 7-point | $2.719e^{-3}$ |
| | | 100 | $3.813e^{-2}$ |
| | | 150 | $9.697e^{-2}$ |
| | | 200 | $2.307e^{-2}$ |
| | Positive part ratio | Overall | $1.158e^{-3}$ |
| | | 50 | $4.979e^{-2}$ |
| | | 75 | $4.724e^{-1}$ |
| | | 75 7-point | $2.944e^{-2}$ |
| | | 100 | $4.979e^{-2}$ |
| | | 150 | $4.412e^{-1}$ |
| | | 200 | $2.231e^{-1}$ |

*Table 6.* Statistical significance results for each difficulty parameter

| $|\mathcal{C}|$ | door | letter | logic | room | social | team |
|---|---|---|---|---|---|---|
| 5 | 94 | 135 | 92 | 86 | 102 | 115 |
| 10 | 139 | 204 | 134 | 129 | 149 | 176 |
| 20 | 258 | 387 | 239 | 240 | 274 | 333 |
| 50 | 618 | 925 | 563 | 578 | 640 | 799 |
| 75 | 920 | 1380 | 841 | 861 | 951 | 1188 |
| 100 | 1218 | 1829 | 1112 | 1140 | 1256 | 1576 |
| 150 | 1844 | 2743 | 1665 | 1729 | 1882 | 2365 |
| 200 | 2470 | 3643 | 2216 | 2317 | 2496 | 3140 |

*Table 7.* Estimated mean token count per prompt ($\lfloor\text{chars}/4\rfloor$) as a function of $|\mathcal{C}|$, for the six template verbalizers.

| Generator | $|\mathcal{C}|$ | Template | LLM | Ratio |
|---|---|---|---|---|
| ImplicationCycle | 5 | 269 | 1 277 | 4.75× |
| | 10 | 340 | 2 138 | 6.28× |
| | 15 | 413 | 3 084 | 7.47× |
| | 20 | 484 | 4 039 | 8.34× |
| EquivalenceCore | 5 | 240 | 945 | 3.94× |
| | 10 | 325 | 1 999 | 6.15× |
| | 15 | 384 | 2 690 | 7.01× |
| | 20 | 469 | 3 766 | 8.03× |
| Backbone | 10 | 312 | 1 989 | 6.38× |
| | 20 | 470 | 3 757 | 7.99× |
| MonoBridge | 10 | 324 | 2 073 | 6.40× |
| | 20 | 469 | 3 900 | 8.32× |
| | 50 | 896 | 9 307 | 10.39× |

*Table 8.* Measured mean prompt-token count (LiteLLM accounting, GPT-OSS-120B tokenizer) for template vs. LLM verbalizations at matched $|\mathcal{C}|$. Only the (generator, $|\mathcal{C}|$) pairs available in both normal and augmented runs are shown.

## P. Tool Calling

Models equipped with code-execution tools have access to off-the-shelf SAT solvers, which leaves open whether the failures documented in Table 1 stem from reasoning competence per se or from the absence of an external solver. To probe this, we evaluate GPT-5.4-mini on 110 items (`logic` verbalizer only) in two configurations: (i) *without tools*; (ii) *with an `execute_python` tool* exposing the `pysat` solver (reasoning disabled, due to an API constraint). The model is free to choose whether to invoke the tool.

**Findings.** (i) Tool calling provides large gains where the bottleneck is search (EquivalenceCore: $+60$ pp; MonoBridge: $+17$ pp), since the model offloads the assignment search to `pysat`. (ii) Tool calling does *not* help on ImplicationCycle ($-2.9$ pp): the model invokes the tool $94\%$ of the time, but the bottleneck lies in correctly translating the natural-language clauses into a CNF the solver can ingest. (iii) Tool calling complements rather than replaces the reasoning evaluation: it isolates the *NL→CNF parsing* step as a residual failure mode, distinct from the search and propagation failures targeted by the main benchmark.

| Model | TP | FP | TN | FN | Acc (%) |
|---|---|---|---|---|---|
| Qwen3-Next-80B | 881 | 44 | 374 | 4 | 96.3 |
| Phi-4-reasoning-plus | 695 | 85 | 135 | 3 | 90.4 |
| GPT-OSS-120B | 877 | 111 | 309 | 21 | 90.0 |
| OLMo-3-32B | 824 | 91 | 327 | 39 | 89.9 |
| Phi-4-reasoning | 804 | 140 | 165 | 14 | 86.3 |
| QwQ-32B | 818 | 143 | 277 | 35 | 86.0 |
| Llama-3.3-70B-Instruct | 886 | 371 | 48 | 13 | 70.9 |

*Table 9.* Per-model confusion matrix for the SAT/UNSAT decision over the full evaluation set. TP/FP/TN/FN are absolute counts; Acc is the overall decision accuracy.

| Generator | $|\mathcal{C}|$ | o3-mini Sat. | o3-mini Wit. | o4-mini Sat. | o4-mini Wit. | GPT-5.4-mini Sat. | GPT-5.4-mini Wit. |
|---|---|---|---|---|---|---|---|
| ImplicationCycle | 5 | 100.0 | — | 98.3 | — | 100.0 | — |
| | 10 | 100.0 | — | 91.7 | — | 91.7 | — |
| | 15 | 96.7 | — | 85.0 | — | 96.7 | — |
| | 20 | 88.3 | — | 81.7 | — | 91.7 | — |
| | 50 | 75.0 | — | 63.3 | — | 65.0 | — |
| | 75 | 50.0 | — | 65.0 | — | 48.3 | — |
| | 100 | 50.0 | — | 60.0 | — | 43.3 | — |
| EquivalenceCore | 10 | 100.0 | 83.3 | 100.0 | 80.0 | 100.0 | 65.0 |
| | 15 | 100.0 | 36.7 | 100.0 | 38.3 | 100.0 | 45.0 |
| | 20 | 100.0 | 40.0 | 100.0 | 48.3 | 98.3 | 36.7 |
| | 50 | 100.0 | 3.3 | 100.0 | 15.0 | 100.0 | 20.0 |
| Backbone | 20 | 100.0 | 100.0 | 100.0 | 100.0 | 100.0 | 100.0 |
| | 50 | 100.0 | 85.0 | 100.0 | 93.3 | 100.0 | 100.0 |
| | 75 | 100.0 | 86.7 | 98.3 | 80.0 | 100.0 | 95.0 |
| | 100 | 100.0 | 78.3 | 100.0 | 86.7 | 100.0 | 95.0 |
| MonoBridge | 10 | 100.0 | 100.0 | 100.0 | 98.3 | 100.0 | 98.3 |
| | 20 | 100.0 | 98.3 | 100.0 | 93.3 | 100.0 | 93.3 |
| | 50 | 100.0 | 81.7 | 100.0 | 75.0 | 98.3 | 86.7 |
| | 75 | 100.0 | 70.0 | 96.7 | 56.7 | 100.0 | 81.7 |
| | 100 | 100.0 | 51.7 | 83.3 | 46.7 | 98.3 | 71.7 |
| | 150 | 100.0 | 40.0 | 53.3 | 8.3 | 90.0 | 38.3 |
| | 200 | 100.0 | 28.3 | 48.3 | 11.7 | 70.0 | 13.3 |

*Table 10.* Decision accuracy (Sat., %) and witness validity (Wit., %, SAT instances only) for three proprietary reasoning models. Each cell averages 10 formulas × 6 template verbalizations, matching the protocol of Table 1. The combined micro-average (witness validity for SAT generators, decision accuracy for ImplicationCycle) is 70.2% (o3-mini), 67.1% (o4-mini) and 71.7% (GPT-5.4-mini).

| Generator | $|\mathcal{C}|$ | Low Sat. | Low Wit. | Medium Sat. | Medium Wit. | High Sat. | High Wit. |
|---|---|---|---|---|---|---|---|
| ImplicationCycle | 5 | 98.3 | — | 91.7 | — | 75.0 | — |
| | 10 | 91.7 | — | 95.0 | — | 80.0 | — |
| | 15 | 76.7 | — | 93.3 | — | 88.3 | — |
| | 20 | 51.7 | — | 93.3 | — | 78.3 | — |
| | 50 | 40.0 | — | 56.7 | — | 30.0 | — |
| | 75 | 21.7 | — | 45.0 | — | 18.3 | — |
| | 100 | 28.3 | — | 40.0 | — | 8.3 | — |
| EquivalenceCore | 10 | 100.0 | 31.7 | 100.0 | 83.3 | 100.0 | 93.3 |
| | 15 | 100.0 | 10.0 | 100.0 | 46.7 | 100.0 | 55.0 |
| | 20 | 98.3 | 6.7 | 100.0 | 50.0 | 98.3 | 63.3 |
| | 50 | 100.0 | 0.0 | 100.0 | 8.3 | 98.3 | 31.7 |
| Backbone | 20 | 100.0 | 100.0 | 100.0 | 96.7 | 100.0 | 100.0 |
| | 50 | 100.0 | 91.7 | 100.0 | 98.3 | 100.0 | 100.0 |
| | 75 | 100.0 | 88.3 | 98.3 | 88.3 | 100.0 | 98.3 |
| | 100 | 100.0 | 78.3 | 100.0 | 83.3 | 100.0 | 95.0 |
| MonoBridge | 10 | 100.0 | 95.0 | 100.0 | 100.0 | 100.0 | 100.0 |
| | 20 | 100.0 | 80.0 | 100.0 | 100.0 | 100.0 | 98.3 |
| | 50 | 86.7 | 51.7 | 100.0 | 83.3 | 100.0 | 93.3 |
| | 75 | 70.0 | 28.3 | 100.0 | 68.3 | 100.0 | 83.3 |
| | 100 | 63.3 | 20.0 | 100.0 | 53.3 | 93.3 | 58.3 |
| | 150 | 53.3 | 6.7 | 90.0 | 31.7 | 81.7 | 35.0 |
| | 200 | 58.3 | 5.0 | 73.3 | 23.3 | 58.3 | 11.7 |

*Table 11.* GPT-OSS-120B decision accuracy (Sat., %) and witness validity (Wit., %, SAT instances only) at three reasoning-effort levels.

| Generator | Without tools | With tools (% code used) |
|---|---|---|
| Backbone | 85.0 | 90.0 (90%) |
| EquivalenceCore | 30.0 | 90.0 (90%) |
| ImplicationCycle | 74.3 | 71.4 (94%) |
| MonoBridge | 82.9 | 100.0 (100%) |
| Overall | 70.9 | 87.3 (95%) |

*Table 12.* GPT-5.4-mini accuracy (%) without vs. with code-execution tool ($n = 110$ items, `logic` verbalizer). The parenthesised number is the proportion of items where the model chose to invoke the tool.

