# OpenReview forum: "Evaluating Robustness of Reasoning Models on Parameterized Logical Problems"
_ICML.cc/2026/Conference — ICML 2026 spotlight_

### Official Review · Reviewer_zNQ2 · 2026-03-11

**Soundness:** 3
**Presentation:** 2
**Significance:** 3
**Originality:** 3
**Overall Recommendation:** 4
**Confidence:** 3

**Summary:**

This paper presents a diagnostic evaluation framework for testing the robustness of large reasoning models on satisfiability problems using parameterized 2-CNF formula families. Rather than treating SAT as a single aggregate benchmark, the paper constructs controlled formula generators that vary specific structural properties—such as contradiction cycles, free-variable multiplicity, planted backbones, late bridge clauses, and duplicated or symmetric patterns—while keeping satisfiability behavior interpretable. It also introduces semantics-preserving perturbations of the same underlying formulas, including changes such as clause reordering, filler clauses, renaming, and repetition, in order to measure whether model performance is stable under superficial transformations. The paper’s main contribution is an evaluation methodology that separates structural reasoning ability from surface sensitivity and provides a more fine-grained view of reasoning robustness than standard SAT accuracy alone.

**Compliance With Llm Reviewing Policy:**

Affirmed.

**Final Justification:**

This paper proposes a useful diagnostic benchmark for evaluating reasoning robustness in controlled 2-CNF settings. Its main strengths are the structured benchmark design and the evaluation beyond aggregate SAT accuracy. My original concerns were mainly about framing and scope rather than technical soundness.  The rebuttal addressed these concerns sufficiently, especially by clarifying the effective sample size, the SAT/UNSAT evaluation protocol, and the exact role of renaming-related experiments. I still think the paper would benefit from slightly tighter framing and a clearer discussion of limitations, but these now seem like revision-level issues. Overall, the rebuttal improved my assessment, and I am raising my score from 3 to 4.

**Key Questions For Authors:**

1. The paper emphasizes variable renaming and structural invariance in the motivation, but the strongest empirical focus is on clause count, filler clauses, clause order, and duplication. Could the authors provide a clearer account of the renaming experiments and how central they are to the main conclusions?

2. How are SAT and UNSAT instances balanced within each generator family and parameter setting, and how sensitive are the main conclusions to that balance?

3. To what extent do the authors intend the conclusions to generalize beyond 2-SAT-style reasoning? In particular, which claims are meant to be benchmark-specific and which are intended as broader claims about LLM/LRM reasoning robustness?

**Limitations:**

No. The authors should explicitly discuss the narrow scope of the benchmark (parameterized 2-CNF rather than broader reasoning tasks), the dependence on templated verbalizations for most main results, the limited number of underlying formulas per setting.

**Strengths And Weaknesses:**

Soundness.
The paper is technically solid as a controlled evaluation study. The benchmark design is careful, the generators are interpretable, and the evaluation goes beyond SAT/UNSAT accuracy by checking witness validity and robustness under semantics-preserving perturbations. The experiments are also reasonably broad across generators, models, clause counts, parameter sweeps, clause order, redundancy, and verbalization styles. That said, the evidence is still narrower than some of the paper’s claims. The study is confined to parameterized 2-CNF families, most results rely on highly templated verbalizations, and each setting uses only 10 underlying formulas with 6 verbalizations per formula, so the amount of truly independent variation is limited. As a result, the paper clearly demonstrates brittleness on this benchmark, but it does not yet justify stronger conclusions about solver-like reasoning more broadly.

Presentation.
The paper is generally clear and easy to follow. The benchmark construction, generator design, difficulty controls, and evaluation protocol are described in enough detail to understand the setup, and the paper is reasonably positioned relative to prior SAT and reasoning-evaluation work. The main weakness is in the framing: the paper sometimes moves too quickly from benchmark-specific findings to broader claims about reasoning competence. The presentation would be stronger if it more clearly separated what is directly supported by the experiments from what is interpretive, and if some details of the experimental protocol were stated more explicitly for reproducibility.

Significance.
Aggregate SAT accuracy is too coarse to diagnose where reasoning models fail, and the proposed benchmark provides a more structured way to study robustness. This is useful for the reasoning-evaluation literature. The impact, however, is narrower than the paper occasionally implies. This is best understood as a diagnostic benchmark for structured logical reasoning under controlled 2-SAT-like settings, not as a broad statement about general reasoning ability.

Originality.
The paper’s originality lies in its evaluation perspective rather than in a new reasoning method. It introduces a controlled diagnostic benchmark built from parameterized 2-CNF families, combines interpretable structural factors with semantics-preserving perturbations, and evaluates models using both decision accuracy and witness validity. This yields a more fine-grained view of reasoning robustness than standard SAT-style benchmarks and helps isolate failure modes that aggregate accuracy would miss. The contribution is therefore meaningful at the level of benchmark design, empirical insight, and problem formulation, even though it does not propose a new solver or learning algorithm.

---

> ### Author Rebuttal · Authors · 2026-03-30
>
> We thank the reviewer for their constructive feedback.
>
> > The paper sometimes moves too quickly from benchmark-specific findings to broader claims about reasoning competence. The presentation would be stronger if it more clearly separated what is directly supported by the experiments from what is interpretive, and if some details of the experimental protocol were stated more explicitly for reproducibility.
>
> We agree. We will revise the introduction and conclusion to explicitly scope our claims to parameterized 2-CNF families and clearly separate empirical observations from interpretive discussion.
>
> > The study is confined to parameterized 2-CNF families, most results rely on highly templated verbalizations, and each setting uses only 10 underlying formulas with 6 verbalizations per formula, so the amount of truly independent variation is limited.
>
> The 6 verbalizations of a given formula are not 6 independent experiments:  they are repeated measurements of the same underlying logical instance, rephrased in different surface forms. We aggregate them per formula before running significance tests. The independent unit is the formula, and we generate 10 per setting, which is the effective sample size.
>
> > The authors should explicitly discuss the narrow scope of the benchmark (parameterized 2-CNF rather than broader reasoning tasks), the dependence on templated verbalizations for most main results, the limited number of underlying formulas per setting.
>
> We will extend the discussion in impact statement.
>
> > The paper emphasizes variable renaming and structural invariance in the motivation, but the strongest empirical focus is on clause count, filler clauses, clause order, and duplication. Could the authors provide a clearer account of the renaming experiments and how central they are to the main conclusions?
>
> Variable renaming is tested through the redundancy/duplication probe (Section 3.5, Section 5.4): each duplicated copy uses a fresh, disjoint set of variable names, so the model encounters the same logical structure under a complete renaming. Additionally, the 6 template verbalizers map variables to different entity names (letters, person names, room numbers, etc.), which constitutes a form of renaming across verbalizations. The similar performance across templates (Figure 8) suggests that models are largely invariant to entity-level renaming within a fixed template structure.
>
> We acknowledge that we do not test renaming *within* a single verbalization (e.g., permuting variable indices while keeping the template fixed). We will clarify which forms of renaming are tested and which are not.
>
> > How are SAT and UNSAT instances balanced within each generator family and parameter setting, and how sensitive are the main conclusions to that balance?
>
> Each generator is either SAT or UNSAT by construction this is a class-level property, not per-instance. ImplicationCycle always produces UNSAT formulas; EquivalenceCore, Backbone, and MonoBridge always produce SAT formulas. Within a given experiment, all instances from the same generator share the same satisfiability status. We therefore do not mix SAT and UNSAT instances within a setting; instead, we evaluate decision accuracy (correctly predicting SAT/UNSAT) and witness validity (for SAT generators only) separately. The SAT bias analysis (Appendix Table 5) measures how often models incorrectly predict SAT on the UNSAT generator.
>
> >To what extent do the authors intend the conclusions to generalize beyond 2-SAT-style reasoning? In particular, which claims are meant to be benchmark-specific and which are intended as broader claims about LLM/LRM reasoning robustness?
>
> Some of our findings are indeed specific to the 2-SAT problem. However, several of our conclusions are intended as broader claims about LLM reasoning, and we believe the evidence supports them at that level: the transitive closure reasoning (multi-hop cause-effect), identifying the pivotal variables (free variables), the sensitivity to non-informative properties, the failure to exploit symmetry, the verbalization sensitivity.

---

> > ### Author Rebuttal · Reviewer_zNQ2 · 2026-04-04
> >
> > Thank you for the thoughtful rebuttal. My main concerns have been adequately addressed. In particular, the clarification on the effective sample size and repeated-measurement design is helpful, the explanation of the SAT/UNSAT protocol is clear, and the clarification of what forms of renaming are and are not tested improves the presentation significantly.
> >
> > I still think the paper would benefit from slightly tighter framing, especially in separating benchmark-specific findings from broader claims about reasoning robustness and in making the current benchmark scope limitations more explicit. However, these now seem like revision suggestions rather than substantive concerns. Overall, the rebuttal has improved my assessment, and I am increasing my score from 3 to 4.

---

> > > ### Author Response · Authors · 2026-04-06
> > >
> > > We thank the reviewer for the careful reading of our rebuttal and for increasing their score. We are pleased that our clarifications addressed your concerns. We also appreciate the suggestion to tighten the framing, and we will incorporate these revisions.

---

### Official Review · Reviewer_goA7 · 2026-03-11

**Soundness:** 3
**Presentation:** 3
**Significance:** 3
**Originality:** 3
**Overall Recommendation:** 5
**Confidence:** 4

**Summary:**

The authors introduce a 2-SAT based benchmark for LLM-based reasoners built from parameterized families of 2-CNFs with controllable implication graph signatures and a variety of other parameters that have no effect on satisfiability but could conceivably increase difficulty for some solvers. They score reasoners not just on whether they correctly identify satisfiability versus unsatisfiability but also on the correctness of their certificates for the same. After CNFs are generated they are converted into an equivalent verbal rendition where each literal is turned into short logical phrase or the negation of said short logical phrase. The authors use their benchmark in an empirical study comparing a family of seven LLM-based reasoners, studying the impact of each difficulty parameter as well as the impact of the parameters that actually have no effect on satisfiability.

**Compliance With Llm Reviewing Policy:**

Affirmed.

**Final Justification:**

The authors have adequately answered my principal concerns regarding the significance of the paper, my questions 2 & 3, and I have therefore upgraded my evaluation of the significance of the paper, as well as my overall recommendation.

**Key Questions For Authors:**

1.	In the sub-section on Difficulty controls, you mention varying “(i) the number of critical variables (e.g., backbone variables)” but you don’t yet define what backbone variables are. I suggest defining them right there, rather than waiting until the Backbone-based Method section, and, moreover, are these the only type of critical variables or are there other types? The use of “e.g.” suggests that there may be others.
2.	In section 3.3 Backbone-based Method, why is the remainder chosen to be monotone? Could this conceivably make the instances easier?
3.	Why did you choose 2-SAT and not Horn-SAT as your underlying problem? At least Horn-SAT is P-complete.
4.	Since your 2-SAT instances have at most 200 clauses, they are solvable by publicly available SAT-solvers. Perhaps you can comment on why you didn’t just choose to generate full 3-SAT instances (perhaps because you wouldn’t be able to use your implication graph methods to control difficulty).

**Limitations:**

Yes

**Strengths And Weaknesses:**

I found the paper to be smoothly written and relatively easy to read. Although I am not at all an expert on 2-SAT, the development of the benchmark seemed principled and sound. That approach is definitely somewhat original – I have not seen a benchmark of this kind before.  I thought the verbalization aspect was a nice twist. Also, the evaluation was quite thorough and relatively interesting. My main reservation about the paper is that I am not very confident about its significance. 2-SAT seems like a very niche problem and is not even P-complete. It would have been just as easy, I believe, to create this same sort of study for the P-complete Horn-SAT problem, and, taking things in a somewhat different direction, for linear programming (also a P-complete problem).

---

> ### Author Rebuttal · Authors · 2026-03-30
>
> We thank the reviewer for their constructive feedback.
>
> > In the sub-section on Difficulty controls, you mention varying “(i) the number of critical variables (e.g., backbone variables)” but you don’t yet define what backbone variables are. I suggest defining them right there, rather than waiting until the Backbone-based Method section, and, moreover, are these the only type of critical variables or are there other types? The use of “e.g.” suggests that there may be others.
>
> Good point. Two of our generators define critical variable types: backbone variables (Backbone, Section 3.3) and free variables (EquivalenceCore, Section 3.2). The other two generators control difficulty through structural parameters: the cycle split position (ImplicationCycle) and the bridge clause position (MonoBridge). We will remove the forward reference "(e.g., backbone variables)" and instead clarify this distinction in the Difficulty Controls subsection.
>
> > In section 3.3 Backbone-based Method, why is the remainder chosen to be monotone? Could this conceivably make the instances easier?
>
> The monotone remainder guarantees satisfiability by construction: if all free variables are set to the same polarity (all true for positive monotone, all false for negative), the remainder is trivially satisfied. This gives us a clean separation between the backbone part (which forces specific variable values via paired clauses) and the remainder (which does not). If the remainder were mixed-polarity, it could introduce additional constraints that interact with the backbone, making it harder to attribute difficulty to a single parameter.
>
> > Why did you choose 2-SAT and not Horn-SAT as your underlying problem? At least Horn-SAT is P-complete.
>
> P-completeness means that Horn-SAT can simulate any polynomial-time computation, but it does not, by itself, provide the fine-grained structural decomposition we need to control difficulty independently of formula size. Indeed, in Horn-SAT, satisfiability is determined by unit propagation (forward chaining), a process that is less amenable to the kind of fine-grained control we exploit. Our choice of 2-SAT was driven primarily by the richness and tractability of its structural theory, which is central to our experimental framework. In particular, implication graph, with its SCC structure, gives us a direct and interpretable mapping from graph-theoretic properties to reasoning difficulty.
>
> We agree that P-completeness is a meaningful complexity criterion, but for our purposes (controlling and explaining difficulty) 2-SAT provides more leverage than Horn-SAT.
>
> > Since your 2-SAT instances have at most 200 clauses, they are solvable by publicly available SAT-solvers. Perhaps you can comment on why you didn’t just choose to generate full 3-SAT instances (perhaps because you wouldn’t be able to use your implication graph methods to control difficulty).
>
> The reviewer is right: our difficulty-control framework depends on the implication graph, which is specific to 2-SAT. If we moved to 3-SAT, we would lose the main tool we use to build instances with specific structural properties. As for the solvability of our instances by SAT solvers: this is intentional. Our goal is to study structural reasoning, not computational hardness. Instances that are easy for solvers but hard for LLMs are exactly what we need to isolate reasoning failures. We will explain this design choice more clearly in the revised paper.

---

> > ### Author Rebuttal · Reviewer_goA7 · 2026-04-02
> >
> > I thank the authors for their well thought-out answers to my questions and have decided to upgrade my evaluation of the paper based on the answers given. I would appreciate it if the authors could integrate their answers to my last two questions into the camera-ready (if the paper is accepted), because I would imagine these questions will be in the minds of many readers.

---

> > > ### Author Response · Authors · 2026-04-06
> > >
> > > We thank the reviewer for the careful reading of our rebuttal and for upgrading their evaluation. We are pleased that our clarifications addressed your concerns. We will incorporate these clarifications into the final version if the paper is accepted.

---

### Official Review · Reviewer_qQWj · 2026-03-12

**Soundness:** 3
**Presentation:** 3
**Significance:** 3
**Originality:** 3
**Overall Recommendation:** 5
**Confidence:** 3

**Summary:**

Main contributions: the paper introduces a novel dataset and experimental framework for evaluating the performance of large reasoning models on logical tasks. The focus is on investigating the brittleness of the models when varying parameters of the generated problems.

A large variety of experiments are performed with different models and parameter sweeps.

The results show strong performance shifts across certain meaning-preserving transformations of the input problems.

**Compliance With Llm Reviewing Policy:**

Affirmed.

**Final Justification:**

This is a well-motivated and timely paper. The rebuttal was very detailed and quite convincing.

**Key Questions For Authors:**

Have you tested out closed-source frontier models like Claude, ChatGPT or Gemini? If so, what did they reveal? If not, please explain why. What about problem classes more difficult than 2-SAT, e.g., 3-SAT? Does the EquivalenceCore test correspond to the free variable stuff mentioned in 3.2?

**Limitations:**

The issue of limitations is mentioned.

**Strengths And Weaknesses:**

The paper is well-written and easy to read. The related work section is comprehensive and the contribution of the paper is made clear.

The experiments are quite convincing and extensive enough. In particular, it is refreshing to see a comprehensive selection of modern LLMs in the experimental section of a paper. The effects of different parameters are analyzed in detail.

Potential weaknesses: The core identified issue of LLMs being sensitive to different meaning-preserving transformations of input prompts (e.g. ordering, wording) is not fully novel. Concerning significance, the paper is of good quality, but focuses on a relatively minor problem concerning LLM reasoning.

Also, it is a small issue, but the impact statement makes the paper go just over 8 pages, so either it should be made part of the Appendix, or the main body of the paper should be condensed to make room for the statement.

---

> ### Author Rebuttal · Authors · 2026-03-30
>
> We thank the reviewer for their constructive feedback.
>
> > The core identified issue of LLMs being sensitive to different meaning-preserving transformations of input prompts (e.g. ordering, wording) is not fully novel. Concerning significance, the paper is of good quality, but focuses on a relatively minor problem concerning LLM reasoning.
>
> We agree that sensitivity to surface form is a known phenomenon. Our contribution is not the observation itself but the *diagnostic methodology*: parameterized 2-CNF families with controllable implication-graph structure that allow attributing failures to specific structural phenomena (cycle length, backbone fraction, bridge position, etc.) rather than to instance size or surface form alone. This level of fine-grained structural control distinguishes our work from prior benchmarks that primarily vary surface realizations of fixed instances.
>
> > the impact statement makes the paper go just over 8 pages, so either it should be made part of the Appendix, or the main body of the paper should be condensed to make room for the statement.
>
> We will move the impact statement in the revised version.
>
> > Have you tested out closed-source frontier models like Claude, ChatGPT or Gemini? If so, what did they reveal? If not, please explain why.
>
> Not in the submitted version, as we focused on open-weight models that we could serve locally for reproducibility and cost control. We have since evaluated o3-mini (see results to reviewer jT3b) and will include these results along with additional proprietary models in the revised paper.
>
> > What about problem classes more difficult than 2-SAT, e.g., 3-SAT?
>
> Moving to 3-SAT raises non-trivial methodological challenges. First, the clean correspondence between problem structure and difficulty (central to our framework) breaks down: while phase transitions near the clause-to-variable ratio threshold (~4.27) provide some handle on hardness, there is no characterization of difficulty comparable to the implication graph for 2-SAT. Second, it becomes difficult to disentangle a model's inability to reason logically from its inability to perform exhaustive search, which is unavoidable for hard 3-SAT instances.
>
> In other words, extending to 3-SAT would introduce a fundamental confound: since 3-SAT is NP-complete, it is unclear whether model failures reflect limited logical reasoning or simply the absence of an efficient solution strategy. Our framework is precisely designed to avoid this ambiguity. That said, we agree that studying harder problem classes is a valuable direction for future work.
>
> > Does the EquivalenceCore test correspond to the free variable stuff mentioned in 3.2?
>
> Yes, exactly. The EquivalenceCore generator (Section 3.2, "SAT Generator via Free Variables") constructs formulas with a set of free variables *F* where each remaining variable is defined by equivalence to a literal over *F*.

---

> > ### Author Rebuttal · Reviewer_qQWj · 2026-04-03
> >
> > I thank the authors for answering all the rebuttal questions in a clear way, also being open about "limitations". In particular, the move to 3-SAT is an interesting threshold, but indeed the reasons for not going there were clearly covered.

---

> > > ### Author Response · Authors · 2026-04-06
> > >
> > > We thank the reviewer for the careful reading and for acknowledging our rebuttal. We are pleased that our clarifications addressed your concerns.

---

### Official Review · Reviewer_jT3b · 2026-03-13

**Soundness:** 3
**Presentation:** 3
**Significance:** 3
**Originality:** 3
**Overall Recommendation:** 4
**Confidence:** 4

**Summary:**

This work presents a novel framework for evaluating LLM reasoning abilities and robustness through language-framed 2-SAT problems. The paper presents several methods to automatically construct both satisfiable and unsatisfiable examples  that have nice heuristic solutions, and show that the LLMs fail to leverage logical observations. These algorithms are controlled by relevant complexity parameters, the effects of which are clearly demonstrated in the paper. The work further ablates different relevant factors, such as the verbalization strategy (how SAT problems become understandable text), or whether the LLMs can leverage repeated patterns. The authors conclude by providing some slightly vague but actionable takeaways from their work, showing current limitations of LLMs.

Overall, I believe that the paper presents an interesting evaluation methodology, which is clearly explained and motivated. In particular, the possibility to construct logically challenging puzzles without overwhelming the model in terms of context seems quite valuable. However, the evaluation can be improved, especially in interpreting where the failure modes arise, and demonstrating that the tasks constitute failures of LLM reasoning and not lack of draft space/context. I will gladly raise my score if the authors address the majority of my concerns.

**Compliance With Llm Reviewing Policy:**

Affirmed.

**Final Justification:**

Following the authors' rebuttal, I have raised my score from 3 to 4. They have included a substantial amount of new evidence that supports the claims of their work and have addressed the majority of my high-level concerns. The authors have demonstrated in the rebuttal that their benchmark remains relevant for some proprietary models and have performed several ablations to show that their findings are supported by their evaluation. Namely, they have shown the benchmark reflects genuine LLM reasoning limitations and not formatting or problem understanding.

I still believe the impact and actionability of the work should be explained in further detail, but given the authors' willingness to engage in the discussion, this seems like a problem that can remain for the next revision of the paper.

**Key Questions For Authors:**

1. Can the authors provide the results in some of their experiments (i.e. Table 1, Figures 1, 6) when binned by the input context length?

2. Similarly, are the discrepancies in the verbalization experiments affected by the induced change in context lenght?

3.  Why did the authors settle on a 16k context length for `Phi-4-reasoning-plus`, when the model's context length is 32k [1]?

4. The paper makes it clear that models fail to approach the 2-SAT tasks, but it is unclear whether that is due to poor implication understanding or brittle logic execution. I believe the authors could address this by performing the following experiments (1 model, 1-2 settings is fine):

    - On unsatisfiable expressions:
        - Provide the precise subset of clauses that form the SCC responsible for the contradiction, stripping away the need to filter distractors. If the model's performance drastically improves, it proves the model is capable of long-range implication tracking, but its primary failure mode is attention. If it still fails, the model possesses a fundamental depth limitation, unable to hold a chain of implications in working memory.
    - On satisfiable expressions:
        - Provide the exact truth values for the foundational variables (those that make the problem challenging, without the filler ones). Because every other variable in these generators is strictly coupled to these root variables, the problem becomes trivially deterministic. Then, if the model succeeds with a near 100% witness validity, it proves the model can understand 2-CNF implications, but lacks the ability to find the critical problem constraints.

5. If possible, could the authors present results on 1 recent stronger proprietary LLM?

6. Can the authors present the confusion matrix on SAT vs UNSAT decisions for each LLM?

7. Please address any outstanding concerns from the **Weaknesses** section.

### References

[1] https://huggingface.co/microsoft/Phi-4-reasoning-plus

**Limitations:**

The authors acknowledge that their work is unlikely to have negative societal impacts, which is a statement I agree with. While they have not specifically detailed their limitations, they have outlined possibilities for future work in the same direction as this project. I believe the pipeline and project are sufficiently self-contained that any potential limitations are relatively insignificant.

**Strengths And Weaknesses:**

## Strengths

The paper is clearly written, conveying its message in a convincing manner. The work is well-motivated, and has produced interesting results on LLM reasoning. First, there is a clear gap between simply guessing the satisfiability of the expression, and the ability of the models to provide a correct witness instance. This effectively penalizes models that guess the correct label without proving it (and the authors uncover a significant validation bias, i.e. with LLaMa-3.3B) Outside of these general impressions, I found the following aspects contribute significantly to the strength of the work:

- **Pipeline Desing**: The pipeline is well thought out, with different experiments catching different failure modes of the LLMs. The parameters are carefully thought out and the paper has analysed a wide range of settings and factors that might impact the given experiment.

- **Code Availability**: The authors have provided the code and configurations they used for generating the evaluations, which enabled me to inspect the efficacy of their pipeline.


## Weaknesses

1. **Broken Evaluations Link**: In the anonymized repository, I was unable to find the datasets used in the paper and preliminary results (due to the `/releases/tag/1.0` folder not existing within the repo). It would be good if the authors fix this during the rebuttal or at least provide some example instances of the verbalized problems.

2. **Difficulty Measurements** (Minor): The parameters the authors define clearly make sense as a factor for difficulty from a human perspective, who is likely to heuristically think about the graph/expression. However, given LLMs can attend on the entire context at the same time, it remains unclear whether the difficulty is caused by an underlying difficulty in the structure, or by providing a harder or longer to understand context. If the authors can include some more information about how $|\mathcal{C}|$ maps to number of input tokens in the different settings, it would nicely address this concern.

3. **Model-related issues**:
    - The limitations of Phi-4 seem to be highly related to the low context length, given that the output for `Phi-4-reasoning-plus` is truncated $30.5\%$ of the time.
    - The authors do not provide the reasoning effort used for `GPT-OSS-120B`, with there  being a very significant difference between `medium` and `high` reasoning.
    - (Minor) The models evaluated are somewhat weak and are all open-source. This prevents the reader from concluding whether the paper describes a fundamental LLM robustness issue or is simply a matter of context insufficiency.

4. (Relatively Minor) It remains unclear how the work can be used to further the development of reasoning models. The benchmark is sensible as a standalone reasoning evaluation, however, it does not convey how the results can impact the future improvement of LLMs (i.e. which aspects are most bottlenecking the reasoning process - memory, logic failure, etc.)

5. **Redundancy Probing**: This method is overall correctly implemented, however, the authors' claim that it checks models' ability to avoid redoing symmetric reasoning is not entirely true. For an algorithm/model to decide to copy the relevant assignment, it must also prove that the 2 (or more) partitions are equivalent, which can arguably be a more complex problem than solving the partitions separately. Furthermore, if the copies are interleaved or shuffled, this task becomes even harder. Therefore, the paper's critique that the models should be able to detect the repeated pattern is unsubstantiated.

---

> ### Author Rebuttal · Authors · 2026-03-30
>
> We thank the reviewer for their constructive feedback.
>
> > Broken Evaluations Link
>
> We apologize — the datasets were hosted as a GitHub Release asset, not mirrored by the anonymization platform. We have now added them directly to the repository (code/dataset_generator/output/dataset.zip). Examples are in Appendix G.
>
> > Difficulty Measurements (Minor) / input length / Binned results?
>
> Each clause maps to ~11 words, so |C| and token count are nearly linear. Two experiments directly control for input length: clause shuffling (identical length, different accuracy) and filler clauses (same core size, added fillers vs more core clauses — different outcomes). The table below shows o3-mini accuracy binned by input token count; while overall accuracy decreases with prompt length, the performance gap between generators persists within each bin, confirming that structural difficulty — not input length alone — drives the results.
>
> | Token bin   | Backbone | EquivCore | ImplCycle | MonoBridge | Overall |
> |-------------|----------|-----------|-----------|------------|---------|
> | 0–256       | —        | —         | 100.0%    | 100.0%     | 100.0%  |
> | 256–512     | 100.0%   | 57.1%     | 94.4%     | 99.0%      | 84.8%   |
> | 512–1024    | 82.3%    | 22.5%     | 65.7%     | 72.5%      | 60.5%   |
> | 1024–2048   | 84.2%    | 0.0%      | 53.6%     | 56.7%      | 60.4%   |
> | 2048–4096   | —        | —         | —         | 37.0%      | 37.0%   |
>
>
> > Phi-4 truncation / Models are weak/open-source / Proprietary LLM?
>
> The 16k output limit for Phi-4 was a GPU compute constraint, causing a 30% truncation rate. We will move Phi-4 to the Appendix. We have already evaluated o3-mini, which achieves 70.2% overall accuracy — higher than all open-weight models on UNSAT (80.0%) but still below 50% on EquivalenceCore (40.8%), confirming that the failures are not limited to open-weight models. Additional proprietary models will be included in the revised paper.
>
> >  reasoning effort used for GPT-OSS-120B
>
> GPT-OSS-120B is an open-weight model that we served locally via vLLM on local HPC, not via the OpenAI API. There is therefore no reasoning effort parameter.
>
> >  development of reasoning models?
>
> Our experiments already identify concrete bottlenecks: distractor sensitivity (fillers degrade performance, Section 5.2), order dependence (shuffling changes accuracy on the same formula, Section 5.3), and shallow decision without constructive reasoning (models predict SAT correctly but fail to produce valid witnesses, Table 1). Each maps to an improvement direction: constraint-filtering training, order-invariant representations, and verifiable certificates. The hint experiments below further disentangle whether the bottleneck is *locating* critical constraints or *executing* implications.
>
> > Redundancy probing claim is unsubstantiated.
>
> We agree; our framing overstated what should be expected. We will revise Section 3.5 to present it as testing *whether models detect and are affected by repeated structure*. Empirically, performance on duplicated formulas (50*2 clauses) is indistinguishable from 100 distinct clauses, with no cross-referencing in reasoning traces — models treat the input as a flat constraint set.
>
> > Verbalization discrepancies affected by context length?
>
> Among the six template verbalizers (Figure 8), performance is very similar despite differences in sentence length, indicating that within a comparable length range surface wording matters little. The LLM verbalizer does produce longer prompts (multi-sentence paragraphs per clause), so the ~25-point drop (Figure 9) may partly reflect increased length. We have measured mean token counts for template vs. LLM verbalizations at matched |C| and confirm a substantial length increase for LLM-generated text. The full comparison will be included in the revised appendix.
>
> > Implication understanding or logic execution?
>
> We **ran two experiments as suggested**:
>
> - **UNSAT core only:** Stripping filler clauses consistently improves accuracy across models (e.g., +5--8 pp for GPT-OSS-120B at 75 clauses), confirming distractor filtering is a bottleneck. But even without fillers, performance stays well below ceiling (~45% for GPT-OSS-120B at 75 clauses), showing implication chain tracking is *also* a bottleneck.
>
> - **SAT hints on foundational variables:** We provided the minimal determining set of variables as hints to GPT-OSS-120B (5700 items). For EquivalenceCore, hints yield +15 to +70 pp (e.g., |C|=5: 30%→100%, |C|=15: 20%→82%, |C|=50: 28%→80%) — the bottleneck is *identifying* free variables, not propagating. For Backbone, hints degrade by 10--25 pp because the large hint list overwhelms the reasoning trace — the model already solves Backbone well without hints.
>
> We will add additional models.
>
> > Confusion matrix for SAT vs UNSAT?
>
> Tab 5 already reports SAT prediction rate on UNSAT instances. We computed the full per-model confusion matrix (TP/FP/TN/FN) and will include.

---

> > ### Author Rebuttal · Reviewer_jT3b · 2026-04-02
> >
> > I thank the authors for the comprehensive response and the new experiments. For the most part, this has resolved my most major concerns and I will raise my score to 4 after the discussion phase concludes. However, the following points still remain unclear/unaddressed in my opinion:
> >
> > 1. "GPT-OSS-120B has no reasoning effort parameter when deployed locally"
> >
> > The model does indeed support different reasoning efforts even when deployed locally. The parameter can be set by feeding `reasoning_effort=<low|medium|high>` when using the tokenizer's `apply_chat_template` function on the standard  conversation format [1]. Though I would agree this is poorly documented, and the default reasoning mode of medium is sufficiently capable. I have personally used this and there is a major difference between the reasoning efforts, even if the model is good in all of them.
> >
> > 2. (New) I am curious whether tool calling models will be able to handle the benchmark much easier given their ability to execute code. This is of course not a critique of the given work, as it tests completely different capabilities, but can be an interesting experiment nonetheless.
> >
> > 3. Why did you consider running o3-mini in your experiments, instead of o4-mini, GPT-5-mini or GPT-5.4-mini, being superior reasoning models of similar cost?
> >
> > If the additional comment size also allows the authors to include tables from the experiments they promised to include in their paper (confusion matrix, and mean token count vs |C| for different verbalizations), that would be great!
> >
> > ### References
> >
> > [1] https://huggingface.co/openai/gpt-oss-120b/discussions/45

---

> > > ### Author Response · Authors · 2026-04-06
> > >
> > > We thank the reviewer for their continued engagement and for raising their score. Below, we address the thoughtful follow-up comments and suggestions, which we will incorporate into the revised version.
> > >
> > > > 1. GPT-OSS-120B reasoning effort
> > >
> > > Our experiments used the default setting (medium). We thank the reviewer for pointing out the reasoning_effort parameter in apply_chat_template. Here are the results with reasoning efforts:
> > >
> > > | | | Low | | Medium | | High | |
> > > |---|---|---|---|---|---|---|---|
> > > | Generator | \|C\| | Sat. | Wit. | Sat. | Wit. | Sat. | Wit. |
> > > | ImplicationCycle | 5 | 98.3 | --- | 91.7 | --- | 75.0 | --- |
> > > | - | 10 | 91.7 | --- | 95.0 | --- | 80.0 | --- |
> > > | - | 15 | 76.7 | --- | 93.3 | --- | 88.3 | --- |
> > > | - | 20 | 51.7 | --- | 93.3 | --- | 78.3 | --- |
> > > | - | 50 | 40.0 | --- | 56.7 | --- | 30.0 | --- |
> > > | - | 75 | 21.7 | --- | 45.0 | --- | 18.3 | --- |
> > > | - | 100 | 28.3 | --- | 40.0 | --- | 8.3 | --- |
> > > | EquivalenceCore | 10 | 100.0 | 31.7 | 100.0 | 83.3 | 100.0 | 93.3 |
> > > | - | 15 | 100.0 | 10.0 | 100.0 | 46.7 | 100.0 | 55.0 |
> > > | - | 20 | 98.3 | 6.7 | 100.0 | 50.0 | 98.3 | 63.3 |
> > > | - | 50 | 100.0 | 0.0 | 100.0 | 8.3 | 98.3 | 31.7 |
> > > | Backbone | 20 | 100.0 | 100.0 | 100.0 | 96.7 | 100.0 | 100.0 |
> > > | - | 50 | 100.0 | 91.7 | 100.0 | 98.3 | 100.0 | 100.0 |
> > > | - | 75 | 100.0 | 88.3 | 98.3 | 88.3 | 100.0 | 98.3 |
> > > | - | 100 | 100.0 | 78.3 | 100.0 | 83.3 | 100.0 | 95.0 |
> > > | MonoBridge | 10 | 100.0 | 95.0 | 100.0 | 100.0 | 100.0 | 100.0 |
> > > | - | 20 | 100.0 | 80.0 | 100.0 | 100.0 | 100.0 | 98.3 |
> > > | - | 50 | 86.7 | 51.7 | 100.0 | 83.3 | 100.0 | 93.3 |
> > > | - | 75 | 70.0 | 28.3 | 100.0 | 68.3 | 100.0 | 83.3 |
> > > | - | 100 | 63.3 | 20.0 | 100.0 | 53.3 | 93.3 | 58.3 |
> > > | - | 150 | 53.3 | 6.7 | 90.0 | 31.7 | 81.7 | 35.0 |
> > > | - | 200 | 58.3 | 5.0 | 73.3 | 23.3 | 58.3 | 11.7 |
> > >
> > > On the SAT generators (EquivalenceCore, Backbone, MonoBridge), high effort consistently improves witness validity over medium. For instance, on EquivalenceCore at |C|=50, witness validity rises from 8.3% (medium) to 31.7% (high), indicating that deeper reasoning helps construct valid assignments. However, on ImplicationCycle (UNSAT), high effort degrades performance relative to medium at nearly every size (e.g., 40.0% → 8.3% at |C|=100). This is interesting as it seems that extended reasoning on UNSAT instances may lead the model into unproductive exploration or over-commitment to candidate assignments rather than identifying the contradiction cycle. We will add and discuss these new results to the paper.
> > >
> > > > 2 (new) Tool calling
> > >
> > > We ran GPT-5.4-mini on 110 items (logic textualizer only), evaluated both without tools (reasoning=medium) and with a Python code execution tool (pysat available, no reasoning due to API limitation).
> > >
> > > | Generator        | Without tools | With tools (used code) |
> > > |------------------|---------------|------------------------|
> > > | Backbone         |        85.0%  |        90.0% (90%)     |
> > > | EquivalenceCore  |        30.0%  |        90.0% (90%)     |
> > > | ImplicationCycle |        74.3%  |        71.4% (94%)     |
> > > | MonoBridge       |        82.9%  |       100.0% (100%)    |
> > > | **Overall**      |    **70.9%**  |    **87.3% (95%)**    |
> > >
> > > Tool calling helps most on EquivalenceCore (+60 pp) and MonoBridge (+17 pp), where the model uses pysat to find satisfying assignments. ImplicationCycle remains difficult even with code — the model must still parse the natural language into clauses correctly. We will include this experiment with a detailed analysis in the revised version.
> > >
> > > > Why o3-mini
> > >
> > > We simply selected o3-mini as a first model. Please find here the results for o4-mini and GPT-5.4-mini.
> > >
> > > | Generator        | o3-mini | o4-mini | GPT-5.4-mini |
> > > |------------------|---------|---------|--------------|
> > > | Backbone         |  87.5%  |  90.0%  |        97.5% |
> > > | EquivalenceCore  |  40.8%  |  45.4%  |        41.7% |
> > > | ImplicationCycle |  80.0%  |  77.9%  |        76.7% |
> > > | MonoBridge       |  67.1%  |  55.7%  |        69.0% |
> > > | **Overall**      |**70.2%**|**67.1%**|    **71.7%** |
> > >
> > > > Requested tables
> > >
> > > Confusion matrix (SAT vs UNSAT decision)
> > >
> > > | Model                  |  TP |  FP |  TN |  FN | Acc   |
> > > |------------------------|-----|-----|-----|-----|-------|
> > > | Qwen3-Next-80B         | 881 |  44 | 374 |   4 | 96.3% |
> > > | GPT-OSS-120B           | 877 | 111 | 309 |  21 | 90.0% |
> > > | OLMo-3-32B             | 824 |  91 | 327 |  39 | 89.9% |
> > > | Phi4-reasoning         | 804 | 140 | 165 |  14 | 86.3% |
> > > | QwQ-32B                | 818 | 143 | 277 |  35 | 86.0% |
> > > | Phi4-reasoning-plus    | 695 |  85 | 135 |   3 | 90.4% |
> > > | Llama-3.3-70B          | 886 | 371 |  48 |  13 | 70.9% |
> > >
> > > Estimated mean token count vs |C| for different verbalizations (chars/4)
> > >
> > > | |C| | door | letter | logic |
> > > |-----|------|--------|-------|
> > > |   5 |   94 |    135 |    92 |
> > > |  20 |  258 |    387 |   239 |
> > > |  50 |  618 |    925 |   563 |
> > > | 100 | 1218 |   1829 |  1112 |
> > > | 200 | 2470 |   3643 |  2216 |
> > >
> > > More details will be integreted to revised version.

---

### Decision · Program_Chairs · 2026-04-30

**Decision:**

Accept (spotlight)

**Comment:**

The paper shows that evaluating LLMs on standard SAT-style benchmarks conflates surface difficulty with the actual structural phenomena that determine satisfiability. To address this, the authors introduce a novel diagnostic framework built from parameterized families of 2-CNF formulas to evaluate structural reasoning and robustness under meaning-preserving perturbations. On the positive side, the reviewers found that the benchmark design is principled, careful, and highly original in how it separates structural reasoning ability from surface-level sensitivity. The reviewers appreciated that the evaluation goes beyond standard aggregate SAT/UNSAT accuracy by rigorously checking witness validity, providing a much more fine-grained view of LLM failure modes. Furthermore, the extensive experiments across various models and parameters were deemed convincing and highly relevant for the reasoning-evaluation community.

The reviewers identified areas of improvement, such as the framing and scope of the paper's broader claims. Specifically, the reviewers suggest tightening the text to explicitly separate benchmark-specific findings regarding 2-SAT problems from the broader claims about general LLM reasoning competence. Additionally, the reviewers recommend expanding the limitations section to explicitly address the narrow scope of parameterized 2-CNFs and the study's reliance on templated verbalizations. I suggest the authors take these, and other comments, into consideration, and then the paper will be better and more polished and higher quality.

We expect the authors to incorporate the feedback from the reviewers, along with all the promised additions and tables from the rebuttal phase, into the final version of the paper.

## Recommendation

Based on these reviews, I recommend accepting this paper. The reviewers were excited about the paper's clear methodology, well-motivated approach, and its capacity to isolate specific reasoning failures that aggregate accuracy metrics typically miss. They also highly commended the authors for their thorough and responsive rebuttal, which included valuable new experiments with frontier proprietary models and tool-calling capabilities. During the discussion, the reviewers also echoed the positive comments. They found that after marinating on the paper and results, they were indeed convinced that this is a very good paper.